# A FRET-based study reveals site-specific regulation of spindle position checkpoint proteins at yeast centrosomes

Yuliya Gryaznova[1†], Ayse Koca Caydasi[1,2†], Gabriele Malengo[3‡], Victor Sourjik[3*‡], Gislene Pereira[1,2*]

[1]DKFZ-ZMBH Alliance, German Cancer Research Centre, Heidelberg, Germany; [2]Centre for Organismal Studies, University of Heidelberg, Heidelberg, Germany; [3]DKFZ-ZMBH Alliance, Centre for Molecular Biology, Heidelberg, Germany

*For correspondence: victor. sourjik@synmikro.mpi-marburg. mpg.de (VS); gislene.pereira@cos. uni-heidelberg.de (GP)

†These authors contributed equally to this work

Present address: ‡Max Planck Institute of Terrestrial Microbiology and LOEWE Research Center for Synthetic Microbiology, Marburg, Germany

Competing interests: The authors declare that no competing interests exist.

**Abstract** The spindle position checkpoint (SPOC) is a spindle pole body (SPB, equivalent of mammalian centrosome) associated surveillance mechanism that halts mitotic exit upon spindle misorientation. Here, we monitored the interaction between SPB proteins and the SPOC component Bfa1 by FRET microscopy. We show that Bfa1 binds to the scaffold-protein Nud1 and the γ-tubulin receptor Spc72. Spindle misalignment specifically disrupts Bfa1-Spc72 interaction by a mechanism that requires the 14-3-3-family protein Bmh1 and the MARK/PAR-kinase Kin4. Dissociation of Bfa1 from Spc72 prevents the inhibitory phosphorylation of Bfa1 by the polo-like kinase Cdc5. We propose Spc72 as a regulatory hub that coordinates the activity of Kin4 and Cdc5 towards Bfa1. In addition, analysis of *spc72Δ* cells shows that a mitotic-exit-promoting dominant signal, which is triggered upon elongation of the spindle into the bud, overrides the SPOC. Our data reinforce the importance of daughter-cell-associated factors and centrosome-based regulations in mitotic exit and SPOC control.

## Introduction

Alongside their canonical role as microtubule organizing centers, centrosomes of metazoans or spindle pole bodies (SPBs, the functional equivalent of the centrosome) of fungi modulate eukaryotic cell division by serving as signaling centers (*Arquint et al., 2014*; *Fu et al., 2015*). In budding yeast, the SPB is associated with components of two linked pathways: the mitotic exit network (MEN) and the spindle position checkpoint (SPOC). The MEN drives mitotic exit (transition from M-G1 phase) after extension of the anaphase spindle into the daughter cell body. The SPOC is a surveillance mechanism that monitors orientation of the mitotic spindle. The SPOC prevents M-G1 transition when the spindle fails to align along the mother to daughter axis and so is unable to deliver one nucleus into the daughter cell (*Bloecher et al., 2000*; *Muhua et al., 1998*; *Pereira et al., 2000*; *Wang et al., 2000*; *Yeh et al., 1995*). SPOC activation inhibits the MEN.

SPBs are embedded in the nuclear envelope throughout the cell cycle and are composed of three distinct layers (inner, central and outer plaques) that are named according to their position with respect to the nuclear envelope (*Jaspersen and Winey, 2004*). SPOC and MEN proteins associate with the cytoplasmic side (outer plaque) of the SPBs, where cytoplasmic microtubules are nucleated. Within this outer plaque, the transforming acidic coiled coil (TACC) family protein Spc72 (the yeast homolog of CDK5RAP2) anchors the γ-tubulin small complexes that nucleate cytoplasmic microtubules (*Knop and Schiebel, 1998*; *Lin et al., 2015*; *Usui et al., 2003*). Nud1 (the yeast homolog of Centriolin) links Spc72 to the central layer of the SPB through Cnm67 - another SPB core protein (*Brachat et al., 1998*; *Elliott et al., 1999*). Nud1 also serves as a scaffold for the recruitment and

**eLife digest** A cell must duplicate its genetic material and then separate the two copies before it divides. This process is carefully controlled so that each new cell receives an identical set of chromosomes after cell division. In budding yeast, new daughter cells grow as a bud on the side a larger mother cell and are eventually pinched off.

A surveillance mechanism in budding yeast monitors the placement of the molecular machine (called the spindle) that separates the copies of the chromosomes. This mechanism then stops the cell from dividing if the spindle is not positioned correctly. Many of the components of this surveillance mechanism – which is called the spindle position checkpoint – associate with structures at the ends of the spindle. However, it was not clear how these components do this and how it helps them to check if the spindle is positioned correctly.

Now, Gryaznova, Caydasi et al. have used a technique called FRET to answer these questions for an important component of the spindle position checkpoint, a protein called Bfa1. The main advantage of FRET is that it can be used to monitor changes in protein-protein interactions in living cells. This approach identified two proteins that provide sites for Bfa1 to bind to at the ends of the spindle. The experiments also showed that Bfa1 specifically detaches from one of these proteins (called Spc72) when the spindle position checkpoint is activated. This action keeps Bfa1 (and therefore the spindle position checkpoint) active, which in turn stops the cell from starting to divide.

Further experiments then showed that Spc72 acts like a regulatory hub that controls Bfa1's activity. This allows an as-yet unidentified mechanism to coordinate cell division with the position of the spindle. The findings of Gryaznova, Caydasi et al. also suggest that unknown factors switch off the spindle position checkpoint when the spindle is correctly positioned to allow the cell to divide. Future work could now aim to identify the mechanism and the unknown factors. Finally, in a related study, Falk et al. show that the spindle position checkpoint is inactivated when one end of the spindle is moved out of the mother cell and into the bud.

activation of MEN kinases (*Gruneberg et al., 2000*; *Rock and Amon, 2011*; *Rock et al., 2013*; *Valerio-Santiago and Monje-Casas, 2011*).

The key element of SPOC is a GTPase activating protein (GAP) complex composed of the GAP Bub2 and its binding partner Bfa1 (*Bardin et al., 2000*; *Pereira et al., 2000*). The GAP-complex inhibits the GTPase Tem1, whose activation would otherwise trigger the MEN (*Bardin et al., 2000*; *Geymonat et al., 2002*; *2003*; *Hu et al., 2001*; *Pereira et al., 2000*; *Scarfone and Piatti, 2015*; *Wang et al., 2000*). This cell cycle arrest provides extra time to enable the cell to correctly align its spindle along the mother-to-daughter cell axis. Loss of SPOC function in cells with misaligned spindles leads to multi-nucleation and anucleation, as cells undergo mitotic exit and cytokinesis irrespective of whether one nucleus has migrated into the daughter cell or not.

The two components of the bi-partite Bfa1-Bub2 GAP complex localize interdependently at the SPB outer plaque throughout the cell cycle (*Lee et al., 2001*; *Pereira et al., 2000*; *2001*). During normal spindle alignment, the Bfa1-Bub2 complex is unequally distributed between the two SPBs, to predominantly associate with the SPB that is directed towards the daughter cell (dSPB) (asymmetric localization) (*Fraschini et al., 2006*; *Pereira et al., 2000*; *2001*). As long as the spindle is correctly positioned, Bfa1-Bub2 remains stably bound to the dSPB (*Caydasi and Pereira, 2009*; *Monje-Casas and Amon, 2009*). As soon as the anaphase spindle moves into the daughter cell compartment, Bfa1 is inactivated by the conserved polo-like kinase Cdc5 that inhibits Bfa1-Bub2 GAP activity (*Geymonat et al., 2002*; *2003*; *Hu and Elledge, 2002*). Upon spindle misalignment, the MARK/PAR family kinase Kin4 is recruited to the Spc72 component of both SPBs (*D'Aquino et al., 2005*; *Maekawa et al., 2007*; *Pereira and Schiebel, 2005*). Phosphorylation of Bfa1 by Kin4 at SPBs converts the asymmetric and stable SPB localization of Bfa1-Bub2 into a symmetric (same amount on each SPB) and dynamic association (*Caydasi and Pereira, 2009*; *Monje-Casas and Amon, 2009*; *Pereira et al., 2000*). Kin4 also reduces the levels of SPB-bound Bfa1 (*Caydasi and Pereira, 2009*). These drastic changes in Bfa1-Bub2 SPB localization are essential for SPOC function and require binding of the 14-3-3 family protein Bmh1 to Bfa1 previously phosphorylated by Kin4 kinase

(*Caydasi et al., 2014*). Although, it is known that Kin4/Bmh1 pathway alters Bfa1-Bub2 association with the SPBs to engage the SPOC, some important questions remain unanswered: How is the Bfa1-Bub2 GAP complex recruited to the SPB outer plaque? How does Bfa1-Bub2 association with the SPB respond to spindle misalignment? Which SPB pool of Bfa1 is regulated by Kin4 and Bmh1?

Here, we have used acceptor photobleaching Förster (fluorescence) resonance energy transfer (FRET) to analyze the interaction between Bfa1-Bub2 and structural components of SPBs. Our data reveal that, in an unperturbed mitosis Bfa1 C-terminus associated with the C-termini of two components of the SPB outer layer, Nud1 and Spc72. SPOC activation specifically disrupted the interaction of Bfa1 with the microtubule linker Spc72 but not with the MEN-scaffolding protein Nud1. The remodeling of Bfa1-Spc72 interaction in response to spindle misalignment required Kin4 and Bmh1. We propose a model in which the Kin4/Bmh1 pathway disturbs the interaction between Bfa1 and Spc72 to prevent the inhibition of the GAP complex by Cdc5. This step is essential for SPOC function and propagates the SPOC signal throughout both cell compartments. Cells lacking *SPC72* were SPOC proficient. However, after prolonged mitotic arrest, we observed that *spc72Δ* cells frequently became binucleated due to SPOC slippage. In binucleated *spc72Δ* cells that possessed two mis-aligned spindles, re-alignment of only one of the two spindles was sufficient to silence the SPOC and promote mitotic exit irrespective of the presence of one mis-placed nucleus. The dominant nature of this impact of one spindle over the other suggests that there is a MEN activating and/or SPOC inhibitory signal that is released upon the passage of one SPB into the bud. These findings highlight both the importance of local regulation at the SPB for SPOC integrity and the key contribution of a daughter-specific mechanism that triggers MEN activation.

## Results

### Acceptor photobleaching FRET identifies protein-protein interactions at the SPB

We employed acceptor photobleaching FRET to assess where Bfa1-Bub2 is located at SPBs. In this technique, protein-protein proximities are determined by the energy transfer between protein pairs labeled with fluorescent FRET donor and acceptor tags. In contrast to the conventional FRET detection, which is based on acceptor emission measurements (i.e., sensitized emission FRET), the acceptor photobleaching FRET monitors the increase in donor fluorescence upon photobleaching of the acceptor (i.e., donor de-quenching) (*Figure 1—figure supplement 1*) (*Llopis et al., 2000*; *Wouters and Bastiaens, 2001*). The proportionate increase in the donor fluorescence intensity after photobleaching of the acceptor directly yields the apparent FRET efficiency ($E_{FRET}$) (*Karpova et al., 2003*; *Kentner and Sourjik, 2009*), which is further corrected for the unspecific signal observed in the donor-only sample (*Figure 1—figure supplement 1B–D*). Detection of this type of FRET signal is more straightforward and more robust than the sensitized emission measurements of FRET that demands multiple corrections for the spectral crosstalk between donor and acceptor fluorophores.

We optimized the bleaching and imaging parameters for acceptor photobleaching FRET by using resident structural SPB proteins as reference associations. The topology of the core SPB proteins including Spc42, Cnm67 and Spc110 has been previously defined by sensitized emission FRET (*Muller et al., 2005*). Based on this analysis, we chose the FRET pairs of Spc42-Cnm67 and Spc110-Cnm67 as a molecule pair that interacted and a molecule pair that did not interact, respectively. As fluorophores, we used monomeric Turquoise (mTUR, donor) and enhanced YFP (EYFP, acceptor). In each case the fluorophore was fused to the C-terminus of each protein (*Goedhart et al., 2010*). In agreement with published data (*Muller et al., 2005*), the acceptor photobleaching FRET technique yielded a positive FRET interaction between Spc42-mTUR and Cnm67-EYFP (*Figure 1—figure supplement 2A*, 14% mean FRET efficiency in the donor-acceptor pair), while no FRET was detected for the Spc110-mTUR and Cnm67-EYFP pair (*Figure 1—figure supplement 2A*). In order to estimate the maximum FRET efficiency that could be obtained for the mTUR-EYFP fluorophore pair at the SPB in our experimental system, we constructed a tandem tag composed of EYFP and mTUR fused to Bfa1. Bfa1-EYFP-mTUR yielded a FRET value of 26% at SPBs (*Figure 1—figure supplement 2B*), to define the maximum value that can be measured for this particular donor-acceptor pair (mTUR-EYFP). We could however detect no FRET in the cytoplasm using the chimeric Bfa1-mTUR-YFP

tandem pair (data not shown), indicating that the concentration of Bfa1 in the cytoplasm sits below our FRET detection limit.

## FRET analysis reveals a close juxtaposition between Bfa1 and both Spc72 and Nud1 at the SPB outer plaque

Immuno-electron microscopy has established that Bfa1-Bub2 localize to the outer plaque (*Pereira et al., 2000*). We therefore analyzed the juxtaposition of Bfa1-Bub2 to SPB outer plaque structural proteins Nud1, Spc72 and Cnm67. C-terminal tagging of Nud1, Spc72 and Cnm67 with fluorescent proteins did not affect their functionality in microtubule organization, as no defect in nuclear migration and positioning was observed in strains encoding tagged proteins (*Figure 1—figure supplement 3A,B*) (*Grava et al., 2006*).

Next, we constructed chromosomally integrated C- or N-terminal fusions of *BFA1* or *BUB2* with *mTUR* or *EYFP* at their respective endogenous loci. The functionality of these gene fusions was confirmed by their ability to maintain a robust SPOC arrest in a *kar9Δ* strain background. Deletion of *KAR9* causes frequent spindle misalignment at non-permissive temperatures (*Miller and Rose, 1998*). In the absence of SPOC function, *kar9Δ* cultures accumulate multi-nucleated or anucleated cells because cells exit mitosis without having segregated chromosomes into the daughter cell compartment (*Bardin et al., 2000*; *Pereira et al., 2000*) (*Figure 1—figure supplement 3C*). *kar9Δ* cells carrying C-terminally tagged *BUB2* or N-terminally tagged *BFA1* were SPOC deficient (*Figure 1—figure supplement 3C*). This indicates that these fusions were not functional and so they were not analyzed further. Cells harboring C-terminal fusions of *BFA1, NUD1* or *SPC72* and N-terminal fusions of *BUB2* with *mTUR* or *EYFP* retained SPOC function (*Figure 1—figure supplement 3C and 3D*).

We analyzed the FRET efficiency of pairings between Bfa1-EYFP and either Nud1-mTUR, Spc72-mTUR or Cnm67-mTUR at the bud-directed SPB in cycling cells (*Figure 1A*). Pairing Bfa1-EYFP with Nud1-mTUR or Spc72-mTUR yielded a FRET signal, whereas no FRET was detected between Bfa1-EYFP and Cnm67-mTUR (*Figure 1A*). Similar FRET efficiencies were measured in metaphase- and anaphase-arrested cells (*Figure 2—figure supplement 1A,B*). Unlike Bfa1, mTUR-Bub2 did not display any FRET when paired with Nud1-EYFP or Spc72-EYFP (*Figure 2—figure supplement 1C*). Importantly, the mTUR-Bub2 and Bfa1-EYFP combination generated a FRET signal at SPBs (*Figure 2—figure supplement 1D*). These data show that the C-terminus of Bfa1 resides in close proximity to the C-termini of both Nud1 and Spc72 at SPBs. The C-terminus of Bfa1 is also positioned in close proximity to the N-terminus of Bub2, in support of their binding to SPBs as a protein complex (*Pereira et al., 2000*).

## Direct physical association of Bfa1 with the C-terminus of Spc72 in vitro

Recombinant proteins have previously been used to demonstrate the direct physical interaction between Nud1 and Bfa1 (*Gruneberg et al., 2000*). To determine whether Bfa1 also directly interacts with Spc72, we used bacterially purified Bfa1 fused to maltose binding protein (MBP-Bfa1), full length Spc72 tagged with glutathione-binding protein (GST) (GST-Spc72) and a C-terminal truncated fragment of Spc72 (codons 231–622) tagged with 6 histidines (6His-Spc72-C, *Figure 1B*). Full length GST-Spc72 bound to MBP-Bfa1 but not to MBP-beads (*Figure 1B*, lanes 1 and 2). The C-terminal 391 amino acids of Spc72 were sufficient to confer this interaction, as 6His-Spc72-C associated with MBP-Bfa1 but not to MBP (*Figure 1B*, lanes 3 and 4). Notably, MBP-Bfa1 or MBP did not interact with an unrelated 6His-tagged protein (6His-Mlc1) to highlight the specific nature of the association with 6His-Spc72-C (*Figure 1B*, lanes 5 and 6). Taken together, Bfa1 directly binds to Nud1 and Spc72 in vitro, suggesting that the interactions established by FRET arise from direct physical interactions.

## Bfa1 interacts with Spc72 and Nud1 at both mSPB and dSPBs

In an unperturbed mitosis, more Bfa1 molecules bind to the daughter directed SPB (dSPB) than to the mother directed SPB (mSPB) (*Pereira et al., 2000*). This behavior is referred to as asymmetric Bfa1 localization. We asked whether differential association of Bfa1 with Nud1 and Spc72 at the two SPBs explains this asymmetric accumulation of Bfa1. To test this hypothesis, we compared FRET for Bfa1-Nud1 and Bfa1-Spc72 pairs at the dSPB and mSPB in metaphase arrested cells where Bfa1 predominately associates with dSPB and is only weakly detectable on the mSPB (*Figure 2A and B*)

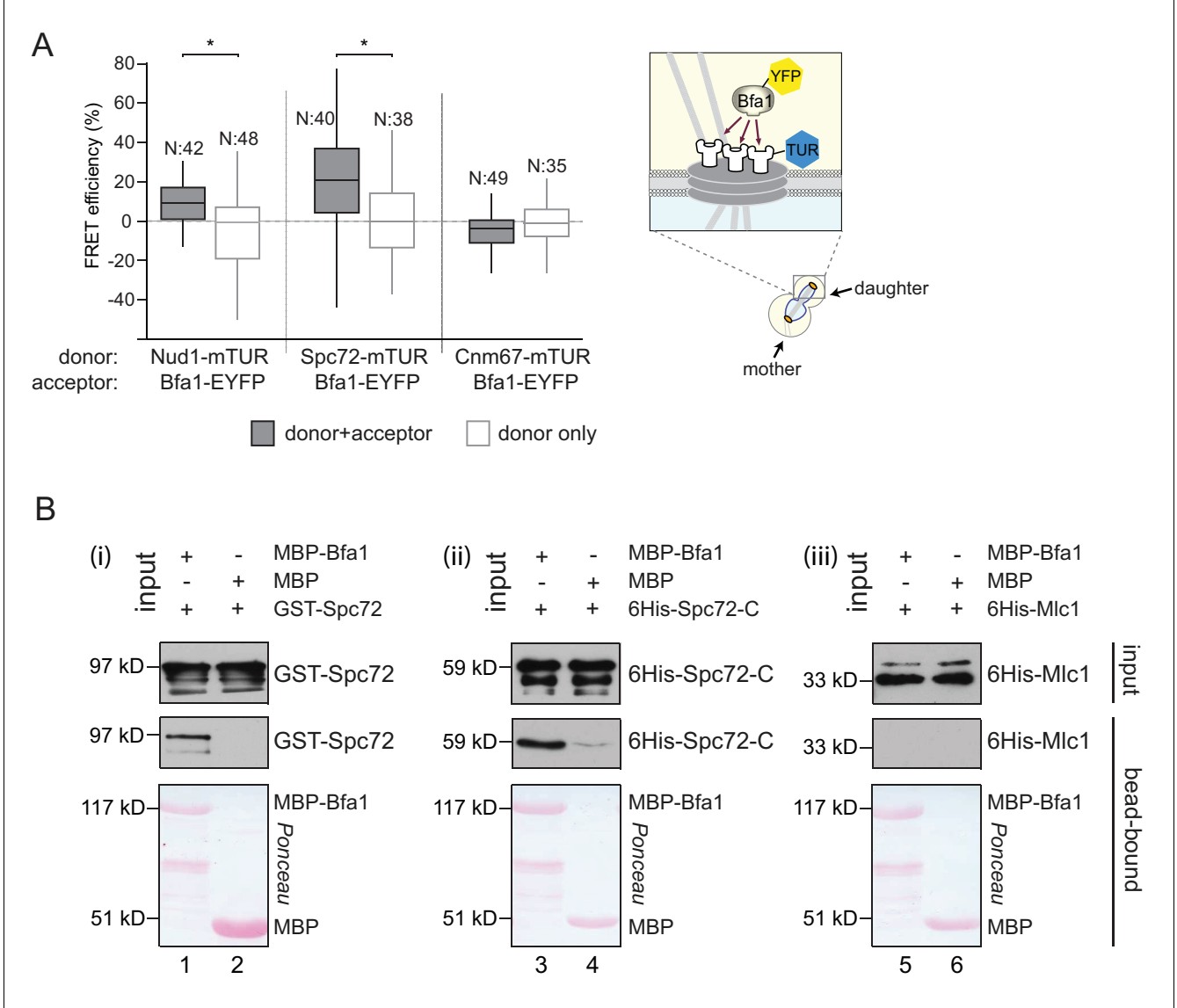

**Figure 1.** Bfa1 interacts with the SPB outer layer proteins Spc72 and Nud1. (**A**) Box-whisker plots representing the distributions of FRET efficiency values for Bfa1 (C-terminally tagged with EYFP) in pair with Nud1, Spc72 or Cnm67 (C-terminally tagged with mTUR) measured at the dSPB as depicted in the cartoon. The FRET data shown here and in subsequent figures are one out of two biological replicates unless otherwise specified. For Box-Whisker plots representing FRET data, the boxes show the lower and upper quartiles, the whiskers show the minimum and maximal values excluding outliers; outliers (not shown) were calculated as values greater or lower than 1.5 times the interquartile range; the line inside the box indicates the median. N is the sample size; asterisks show significant difference according to student's t-test (p<0.01) and the exact p-values are indicated in the accompanying data file (*Figure 1—source data 1*). (**B**) In vitro binding assay of bead bound recombinant MBP-Bfa1 with bacterially purified GST-Spc72 (i), 6His-Spc72-C (codons 231–622) (ii) and 6His-Mlc1 (iii). MBP on beads and 6His-Mlc1 were used as negative controls for in vitro binding reaction. Spc72 was detected with anti-GST antibody; Spc72-C and Mlc1 were detected with anti-6His antibodies after immunoblotting. The Ponceau S stained membrane shows the levels of MBP-Bfa1 and MBP used in the assay. One representative blot out of two independent experiments is shown in each panel.

The following source data and figure supplements are available for figure 1:

**Source data 1.** Raw data and the calculated FRET efficiencies of Nud1-Bfa1 and Spc72-Bfa1 pairs at SPBs in cycling cells (source data for *Figure 1A*).

**Figure supplement 1.** The basic principles of acceptor photobleaching technique to measure FRET.

**Figure supplement 2.** Validation of acceptor photobleaching technique with FRET-positive and FRET-negative controls.

*Figure 1 continued on next page*

Figure 1 continued

**Figure supplement 2—source data 1.** Raw and calculated FRET efficiencies of Spc42-Cnm67 and Spc110-Cnm67 pairs at SPBs in cycling cells (source data for *Figure 1—figure supplement 2A*).
**Figure supplement 2—source data 2.** Raw and calculated FRET efficiencies of Bfa1-EYFP-mTUR tandem pair at SPBs in cycling cells (source data for *Figure 1—figure supplement 2B*).
**Figure supplement 3.** Functionality of tagged SPB proteins and SPOC components.

(*Pereira et al., 2000*). The metaphase arrest was achieved through the depletion of the anaphase promoting complex regulatory subunit Cdc20 (*Shirayama et al., 1999*). The levels of FRET between Bfa1-Nud1 or Bfa1-Spc72 were similar at the mSPB and dSPB (*Figure 2C and D*). This indicates that Bfa1 is in close proximity with Nud1 and Spc72 on both SPBs. Thus, asymmetric SPB localization of Bfa1 cannot be dictated by the differential association of Bfa1 with Spc72 or Nud1.

Importantly, FRET efficiency at SPBs was estimated based on the relative change in the fluorescence of the donors Nud1-mTUR or Spc72-mTUR, both of which localize at both SPBs to similar levels (*Erlemann et al., 2012*; *Knop and Schiebel, 1997*; *1998*). However, the acceptor (Bfa1-EYFP) is recruited to higher levels at the dSPB than at the mSPB to generate different donor:acceptor ratios on the dSPB and mSPB (*Figure 2C and D*). To evaluate the effect of different donor:acceptor ratios on FRET efficiency, we compared the FRET efficiency of Bfa1-Nud1 or Bfa1-Spc72 pairs with the fluorescence signal intensity of the acceptor (Bfa1-EYFP) before bleaching (*Figure 2E and F*). The values of FRET efficiency did not correlate with the signal intensity of the acceptor. Thus, changes in the donor:acceptor ratio do not affect FRET measurements in our experimental system. This validates the ability of this approach to compare the association of Bfa1 with Nud1 and with Spc72 on different poles.

## Kin4/Bmh1 specifically decreases the Bfa1-Spc72 association at SPBs in response to spindle alignment defects

Phosphorylation of Bfa1 by Kin4 kinase decreases the residence time of Bfa1 at SPBs (*Caydasi and Pereira, 2009*). We therefore asked whether Kin4 affected the association of Bfa1 with Nud1, Spc72 or both partners. To this end, we first analyzed cells overproducing Kin4 (*Figure 3A and B*). Overexpression of *KIN4* activates the SPOC to arrest cells in late anaphase, even when the spindle is correctly aligned (*D'Aquino et al., 2005*). As a control for cells that arrest in late anaphase without SPOC activation, we overexpressed a non-degradable version of the mitotic cyclin *CLB2 (clb2ΔDB)* (*Surana et al., 1993*) (*Figure 3A and B*). FRET between Bfa1 and Nud1 was maintained in *KIN4* overexpressing cells (*Figure 3A*, *Figure 3—figure supplement 1*). In contrast, the FRET between Bfa1-Spc72 pair disappeared upon *KIN4* overexpression (*Figure 3B*). Thus, Kin4 activity specifically disturbs the interaction between Bfa1 and Spc72 but not that between Bfa1 and Nud1.

We previously described how binding of Bmh1 to Kin4-phosphorylated Bfa1 abolishes the stable association of Bfa1 with SPBs (*Caydasi et al., 2014*). In cells lacking *BMH1*, Bfa1 remains asymmetrically and stably associated with SPBs, even when Kin4 phosphorylates Bfa1 (*Caydasi et al., 2014*). We therefore proposed that Bmh1 would be required to break the Bfa1-Spc72 interaction. To test this notion, we compared the FRET efficiency between Bfa1 and Spc72 when Kin4 was overproduced in wild type and in *bmh1Δ* cells (*Figure 3C*, *Figure 3—figure supplement 1*). Kin4 overexpression in the presence of *BMH1* greatly decreased the FRET efficiency of the Bfa1-Spc72 pair (*Figure 3C*). This was not the case in *bmh1Δ* cells (*Figure 3C*). Deletion of *BMH1* did not influence the FRET efficiency of Nud1–Bfa1 pair (*Figure 3D*).

To determine whether the interaction between Bfa1 and Spc72 is lost upon SPOC activation, we analyzed the FRET efficiencies of Bfa1-Nud1 and Bfa1-Spc72 in cells with correctly and incorrectly positioned anaphase spindles (*Figure 3E and F*). Spindle position had no impact upon FRET between Bfa1 and Nud1 (*Figure 3E*). In contrast, spindle misalignment reduced the FRET efficiency of the Bfa1-Spc72 pair to the level seen in the donor-only control (*Figure 3F*). In the absence of *KIN4*, however, the FRET between Bfa1 and Spc72 persisted irrespective of spindle position (*Figure 3G and H*). Together, these data show that, upon spindle misalignment, Kin4, with the

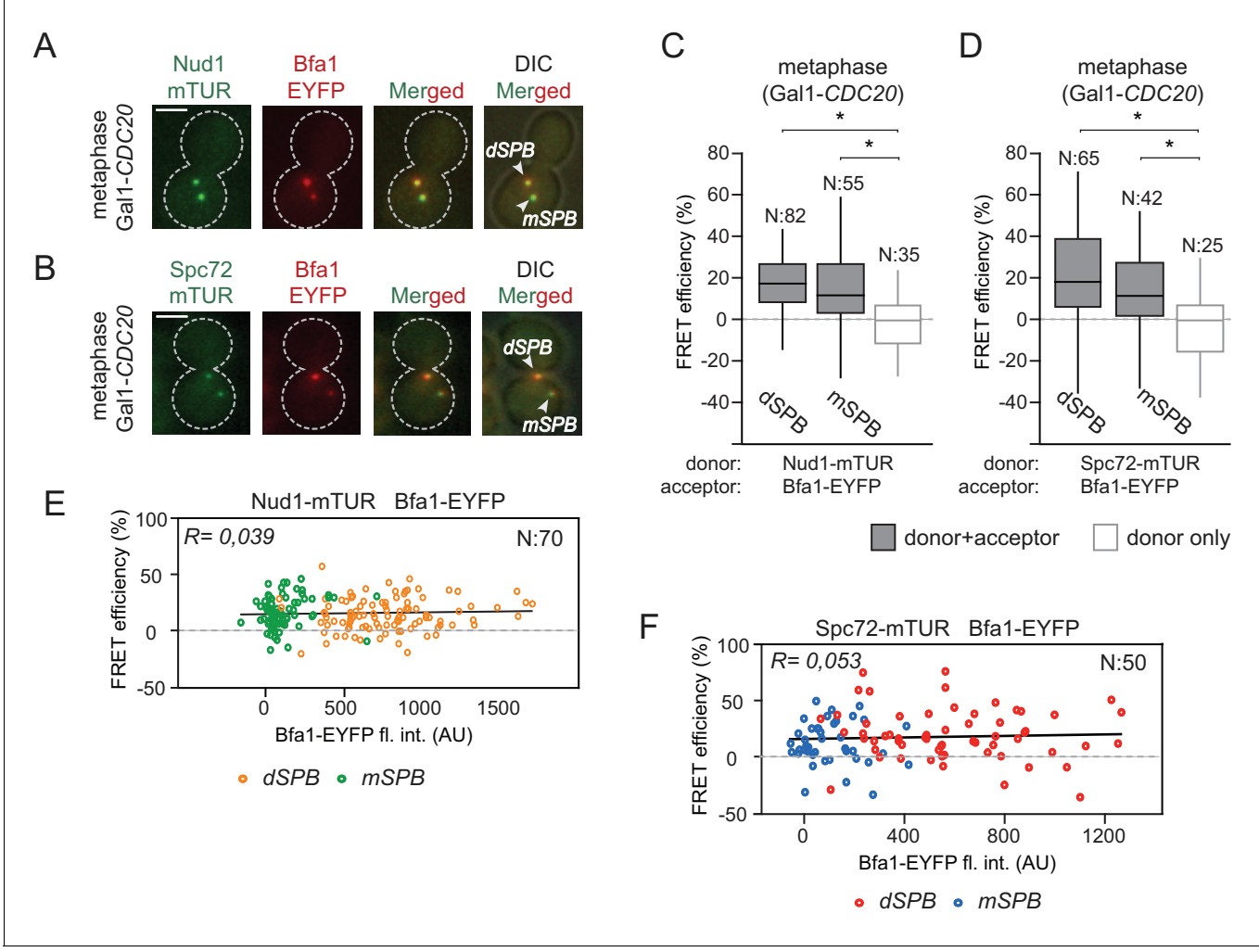

**Figure 2.** Bfa1 interacts with Spc72 and Nud1 at the daughter and mother SPBs. (A–B) Representative images of metaphase-arrested cells carrying Nud1-mTUR Bfa1-EYFP (A) and Spc72-mTUR Bfa1-EYFP (B). Scale bar: 3 μm. (C–D) Box and Whiskers plots showing FRET efficiency for the indicated Nud1-Bfa1 (C) and Spc72-Bfa1 (D) pairs at the daughter and mother SPBs (dSPB and mSPB) in metaphase-arrested cells. N: sample size. Asterisks show significant difference according to student's t-test (p<0.01). See the accompanying data files (*Figure 2—source data 1* and *2*) for exact p-values. (E–F) Scatter plots of FRET efficiencies for Nud1-Bfa1 (E) and Spc72-Bfa1 (F) pairs as a function of Bfa1-EYFP fluorescence intensity values obtained from the dSPB and from the mSPB. N: Sample size. R: r-squared value for the best-fit trendline.

The following source data and figure supplements are available for figure 2:

**Source data 1.** Raw data and the calculated FRET efficiencies of Nud1-Bfa1 and Spc72-Bfa1 pairs at SPBs in metaphase arrested cells (source data for *Figure 2C*).

**Source data 2.** Raw data and the calculated FRET efficiencies of the Spc72-Bfa1 pair at the mother and the daughter SPB in metaphase arrested cells (source data for *Figure 2D*).

**Figure supplement 1.** FRET analysis of Bfa1 and Bub2.

**Figure supplement 1—source data 1.** Raw and calculated FRET efficiencies of the Bfa1-Nud1 pair in metaphase and anaphase arrested cells (source data for *Figure 2—figure supplement 1A*).

**Figure supplement 1—source data 2.** Raw and calculated FRET efficiencies of the Bfa1-Spc72 pair in metaphase and anaphase arrested cells (source data for *Figure 2—figure supplement 1B*).

*Figure 2 continued on next page*

*Figure 2 continued*

**Figure supplement 1—source data 3.** Raw and calculated FRET efficiencies of Bub2-Nud1 and Bub2-Spc72 pairs in cycling cells (source data for *Figure 2—figure supplement 1C*).

**Figure supplement 1—source data 4.** Raw and calculated FRET efficiencies of Bub2-Bfa1 pair in cycling cells (source data for *Figure 2—figure supplement 1D*).

assistance of Bmh1, rearranges Bfa1 molecules on the SPB by specifically influencing the Bfa1-Spc72 interaction.

## Bfa1-SPB binding in the absence of Spc72

Our FRET data shows that SPOC activation specifically interrupts Bfa1 association with Spc72 but not with Nud1 (*Figure 3E and F*). We therefore asked whether there were two separate pools of Bfa1, one that binds Spc72 and another Nud1 (*Figure 4A–i*) or whether Bfa1 molecules bind both Spc72 and Nud1 simultaneously to form a single pool (*Figure 4A-ii*). In the scenario where two individual Bfa1 pools exist (*Figure 4A–i*), Spc72 should recruit the majority of Bfa1 to SPBs because SPB-bound Bfa1 levels fall drastically upon spindle misalignment (*Caydasi and Pereira, 2009*). This coincides well with the loss of Bfa1-Spc72 interaction observed by FRET. Therefore, if two separate Spc72- and Nud1-bound Bfa1 pools exist, less Bfa1 would bind to SPBs in the absence of *SPC72* (*Figure 4A–i*). We therefore analyzed Bfa1 localization in cells lacking *SPC72*. For this analysis, we used the W303 strain background in which *SPC72* is not essential (*Hoepfner et al., 2002*; *Soues and Adams, 1998*) yet Bfa1-Bub2 SPB binding and phospho-regulation is the same as in S288C background where *SPC72* is essential (*Caydasi and Pereira, 2009*; *D'Aquino et al., 2005*; *Monje-Casas and Amon, 2009*; *Pereira and Schiebel, 2005*). Cells lacking *SPC72* cannot form long cytoplasmic microtubules; hence logarithmically growing *spc72Δ* cultures contain cells with misaligned spindles alongside cells with correctly aligned spindles (*Hoepfner et al., 2002*; *Soues and Adams, 1998*). FRET analysis showed that Bfa1 associated with Nud1 at SPBs of *spc72Δ* cells regardless of spindle orientation (*Figure 4B and C*). Interestingly, SPB-bound Bfa1 levels were not reduced in *spc72Δ* cells (*Figure 4D*), arguing against the model where there are two pools of Bfa1 that separately bind Spc72 and Nud1.

To gain deeper insight into how Bfa1 associates with SPBs in the absence of Spc72, we used FRAP (Fluorescence Recovery After Photobleaching) analysis to compare the dynamics of Bfa1 association with SPBs in *spc72Δ* cells with normal and mis-aligned spindles. After photobleaching, Bfa1-GFP failed to recover at SPBs in cells with normally aligned anaphase spindles in the presence of *SPC72* (*Figure 4E*). This confirms the previously reported immobile pool of Bfa1 at the dSPB (*Caydasi and Pereira, 2009*; *Monje-Casas and Amon, 2009*). Cells lacking *SPC72* also recruited Bfa1-GFP stably to the dSPB when the anaphase spindle was correctly aligned (*Figure 4E*). Thus, Bfa1 interaction with Nud1 is stable throughout anaphase when spindle is correctly aligned, independently of Spc72. Next, we examined cells with misaligned spindles. Consistent with previous data, in the presence of *SPC72*, Bfa1 was highly dynamic at both SPBs with a half recovery time of ~ 11 s during spindle misalignment (*Figure 4F and 4G*) (*Caydasi and Pereira, 2009*). The half recovery time of Bfa1 was 6 times longer in *spc72Δ* cells than in *SPC72* cells (*Figure 4F and 4G*). These data suggest that Spc72 regulates the mode of Bfa1 association with Nud1 when the spindle is mispositioned.

Together, our data favor a model in which Bfa1 molecules simultaneously associate with Nud1 and Spc72 (*Figure 4A, ii*). In cells with properly aligned spindles, Nud1 is sufficient to stably recruit Bfa1 to SPBs independently of Spc72. However, in cells with misaligned spindles, Spc72 becomes indispensable to regulate Bfa1 SPB binding dynamics.

## The SPOC is functional in *spc72Δ* cells

Bfa1-SPB binding dynamics in *spc72Δ* cells during spindle misalignment were reminiscent of those seen in *kin4Δ* cells (*Figure 4F and G*). This is consistent with previous reports that Spc72 recruits Kin4 to SPBs and that Kin4 SPB binding is necessary for its function in SPOC (*D'Aquino et al., 2005*;

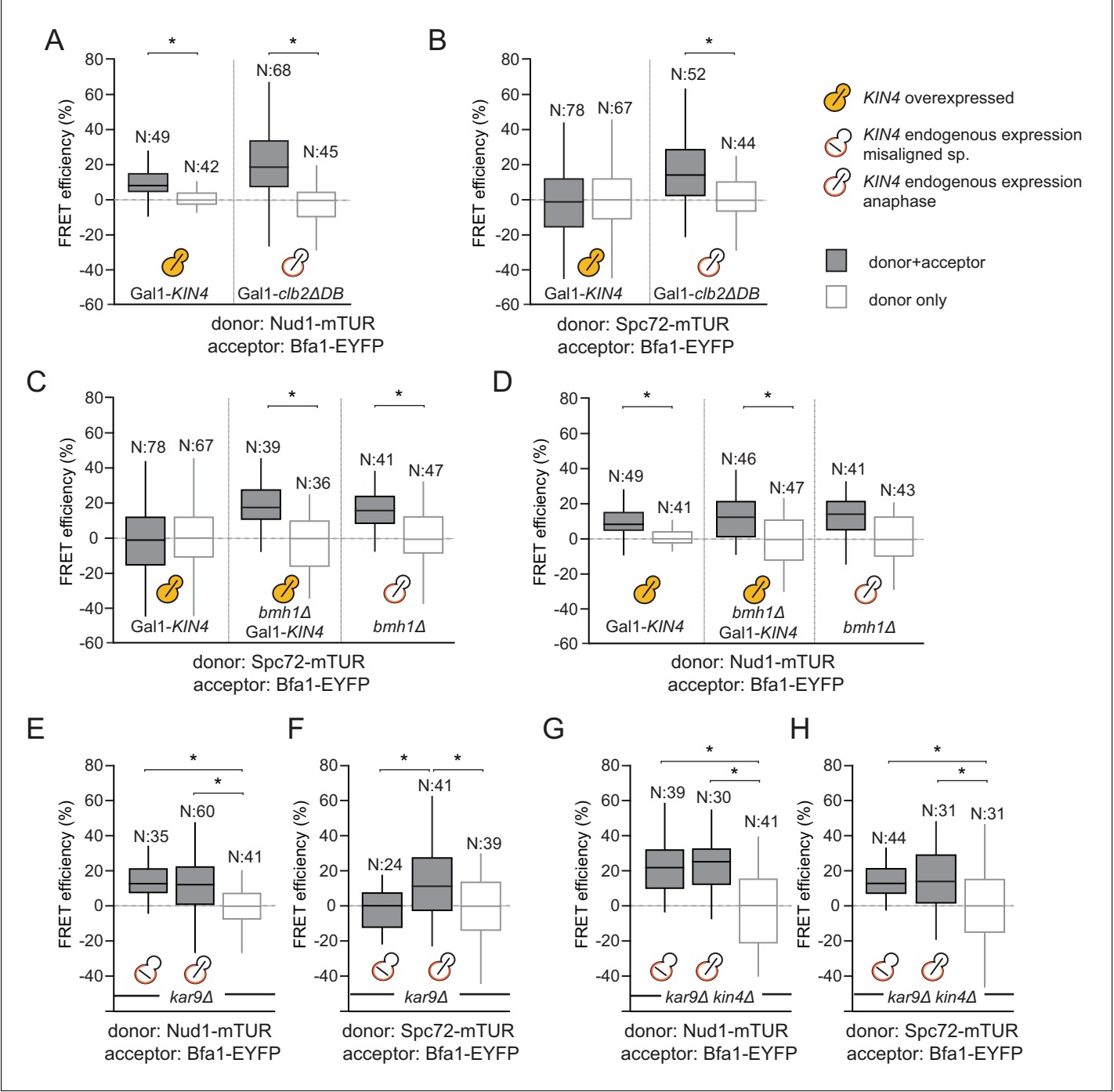

**Figure 3.** SPOC activation interferes with Bfa1-Spc72 interaction. (**A–B**) FRET efficiencies for Nud1-Bfa1 (**A**) and Spc72-Bfa1 (**B**) pairs. Cells were arrested in anaphase by *clb2ΔDB* overexpression (Gal1-*clb2ΔDB*) or by *KIN4* overexpression (Gal1-*KIN4*). (**C–D**) FRET efficiencies for Spc72-Bfa1 (**C**) and Nud1-Bfa1 (**D**) pairs in Gal1-*KIN4* (please note that the data for Gal1-*KIN4* cells is identical to *Figure 3A*), Gal1-*KIN4 bmh1Δ* and *bmh1Δ* cells grown in galactose medium. (**E–F**) FRET efficiencies for Nud1-Bfa1 (**E**) and Spc72-Bfa1 (**F**) pairs in *kar9Δ* cells with correct and mis-aligned spindles. (**G–H**) FRET efficiencies of Nud1-Bfa1 (**G**) and Spc72-Bfa1 (**H**) pairs in *kar9Δ kin4Δ* cells with correct and mis-aligned spindles. Box-whisker plots in G and H are one out of two technical replicates from the same experiment. N: sample size. Asterisks show significant difference according to student's t-test (p<0.01). See the accompanying data files for exact p-values (*Figure 3—source data 1–8*).

The following source data and figure supplement are available for figure 3:

**Source data 1.** Raw data and the calculated FRET efficiencies of Nud1-Bfa1 and Spc72-Bfa1 pairs at SPBs upon *KIN4* overexpression (source data for *Figure 3A*).

*Figure 3 continued on next page*

*Figure 3 continued*
**Source data 2.** Raw data and the calculated FRET efficiencies of the Spc72-Bfa1 pair at SPBs upon *KIN4* overexpression, and *CDC20* depletion (source data for *Figure 3B*).
**Source data 3.** Raw data and the calculated FRET efficiencies of the Spc72-Bfa1 pair in the presence and absence of *BMH1* (source data for *Figure 3C*).
**Source data 4.** Raw data and the calculated FRET efficiencies of the Nud1-Bfa1 pair in the presence and absence of *BMH1* (source data for *Figure 3D*).
**Source data 5.** Raw data and the calculated FRET efficiencies of the Nud1-Bfa1 pair in *kar9Δ* cells with normally aligned or misaligned spindles (source data for *Figure 3E*).
**Source data 6.** Raw data and the calculated FRET efficiencies of the Spc72-Bfa1 pair in *kar9Δ* cells with normally aligned or misaligned spindles (source data for *Figure 3F*).
**Source data 7.** Raw data and the calculated FRET efficiencies of the Nud1-Bfa1 pair in *kar9Δ kin4Δ* cells with normally aligned or misaligned spindles (source data for *Figure 3G*).
**Source data 8.** Raw data and the calculated FRET efficiencies of the Spc72-Bfa1 pair in *kar9Δ kin4Δ* cells with normally aligned or misaligned spindles (source data for *Figure 3H*).
**Figure supplement 1.** Controls for Kin4 overproducing cells used in FRET experiments.

*Maekawa et al., 2007*; *Pereira and Schiebel, 2005*). Indeed, we were unable to detect Kin4 at SPBs in *spc72Δ* cells by fluorescence microscopy (*Figure 5A and B*). Neither could phospho-specific antibodies detect Bfa1 phosphorylation at S180 (one of the two sites phosphorylated by Kin4) in *spc72Δ* cells (*Maekawa et al., 2007*) (*Figure 5C*). Furthermore, the failure of Bfa1 to bind symmetrically (with same levels) to SPBs in *spc72Δ* cells upon spindle misalignment was also reminiscent of the *kin4Δ* phenotype (*Figure 5D*). Collectively, these data indicate that Kin4 is unable to phosphorylate Bfa1 and dislodge Bfa1 from SPBs in the absence of Spc72. Thus, the reduction of Bfa1-SPB binding dynamics in *spc72Δ* cells with misaligned spindles is likely to arise from a lack of Kin4 phosphorylation of Bfa1.

We next asked whether the SPOC is functional in *spc72Δ* cells. For this, we performed live cell imaging of *spc72Δ* and *kar9Δ* cells carrying tubulin tagged with GFP (*GFP-TUB1*) to decorate microtubules (*Figure 6*). We followed cells in which the mitotic spindle either elongated along the mother to daughter cell axis (*Figure 6A–C*, correct spindle orientation, upper rows) or within the mother cell compartment (*Figure 6A–C*, misaligned spindle, lower rows). We defined the duration of anaphase as being the time that elapsed between the onset of rapid spindle elongation phase and spindle breakdown (which is a consequence of mitotic exit) (*Figure 6D*) (*Bardin and Amon, 2001*). As previously reported (*Bloecher et al., 2000*; *Pereira et al., 2000*), *kar9Δ* cells with mis-oriented spindles failed to exit mitosis and displayed a prolonged anaphase arrest indicative of SPOC proficiency (*Figure 6A and D*). Deletion of *KIN4* in *kar9Δ* cells induced cells with misaligned and properly aligned spindles to exit mitosis with similar timing (*Figure 6B and D*), reflecting the SPOC deficiency of *kin4Δ* cells. To our surprise, *spc72Δ* cells were able to sustain an anaphase arrest for the duration of the time-lapse movie (>60 min) upon spindle mis-orientation (*Figure 6C and D*). This delay did not arise from a general deficiency that would extend anaphase because cells exited mitosis 23 min after the onset of spindle elongation when their spindle elongated along the correct mother-daughter cell axis (*Figure 6C and D*). Furthermore, deletion of *BFA1* was lethal in *spc72Δ* cells (*Figure 6— figure supplement 1*), indicating that these cells require SPOC function for survival. Thus, despite the loss of Bfa1 phosphorylation by Kin4, disruption of Bfa1-Spc72 interaction by means of *SPC72* deletion did not compromise the ability of cells with misaligned spindles to engage SPOC arrest.

In wild type cells, Bfa1 phosphorylation by Kin4 is required to prevent Cdc5 phosphorylation of Bfa1, which would otherwise inhibit Bfa1-Bub2 GAP activity and so promote mitotic exit (*D'Aquino et al., 2005*; *Geymonat et al., 2003*; *Pereira and Schiebel, 2005*). We therefore asked whether Cdc5 phosphorylation of Bfa1 is blocked in the absence of *SPC72*. In order to examine Bfa1 phosphorylation by Cdc5, we assessed Bfa1 migration on SDS page gels. Bfa1 becomes hyper-

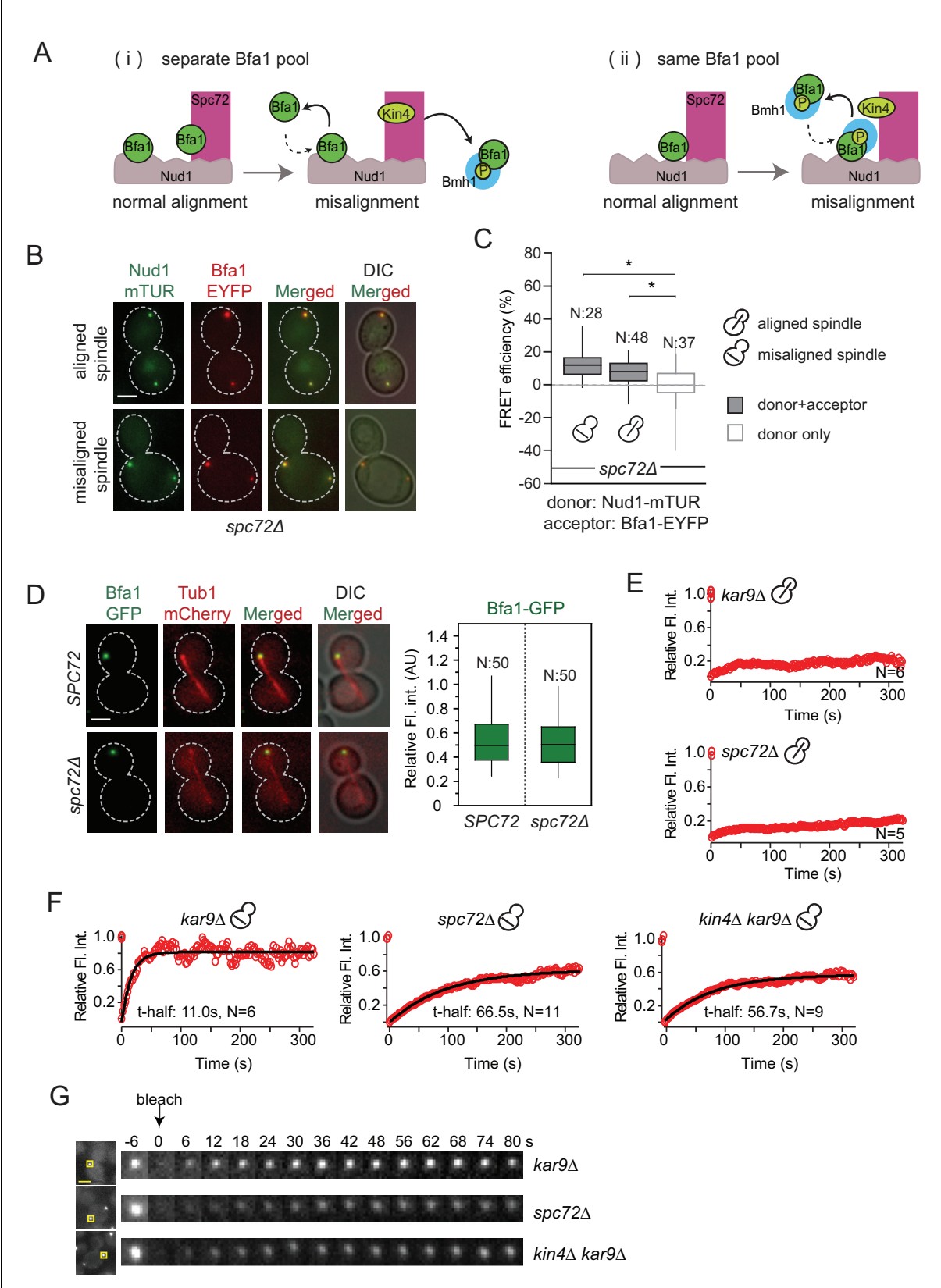

**Figure 4.** Bfa1-SPB binding dynamics in the absence of Spc72. (**A**) Schematic representation of two possible mechanisms for Bfa1 binding at the SPB. i: Bfa1 is recruited as two separate pools. ii: Bfa1 associates simultaneously with Nud1 and Spc72. Dashed arrows indicate dynamic binding. (**B–C**)
*Figure 4 continued on next page*

*Figure 4 continued*

Representative still images and FRET efficiencies for *spc72Δ NUD1-mTUR BFA1-EYFP* cells with aligned and misaligned spindle. Box-whisker plot is one out of two technical replicates from the same experiment. N: sample size. Asterisks show significant difference according to student's t-test (p<0.01). See the accompanying data file for exact p-values (*Figure 4—source data 1*). (D) Representative still images and mean fluorescence intensities of Bfa1-GFP localized at the dSPB during anaphase in *SPC72* (pRS-*SPC72 spc72Δ*) and *spc72Δ* cells carrying *mCherry-TUB1*. Box-whisker plot is one out of two technical replicates from the same experiment. See the source file (*Figure 4—source data 2*) for the raw data. (E-F) FRAP analysis of Bfa1-GFP at the SPBs in cells with correctly aligned (E) and misaligned spindles (F). The black line depicts the best-fit single exponential curve for each data set. Data represent the mean of "N" sized sample. t-half: half recovery time. The graphs show the average fluorescence recovery curves for the corresponding strains. See the accompanying data file for individual curves and raw data (*Figure 4—source data 3* and *4*). Data represented in E is one out of two biological replicates. Data for *spc72Δ* in F is one out of two biological replicates. Data for *kar9Δ* and *kin4Δ kar9Δ* comes from one experiment, whose results are in concordance with published data (*Caydasi et al., 2014*; *Caydasi and Pereira, 2009*). (G) Representative still images of (F). Photobleached SPB is marked with squares. Time-lapse series show 3-fold enlarged photobleached regions at the indicated time points. Time zero is the first image taken after photobleaching. Scale bar: 3 μm.

The following source data is available for figure 4:

**Source data 1.** Raw data and the calculated FRET efficiencies of the Nud1-Bfa1 pair in *spc72Δ* cells with normally aligned or misaligned spindles (source data for *Figure 4C*).
**Source data 2.** Raw and normalized mean fluorescence intensities of Bfa1-GFP at SPBs of *spc72Δ* and *SPC72* cells (source data for *Figure 4D*).
**Source data 3.** Raw and normalized FRAP data of Bfa1-GFP at the SPBs of *spc72Δ* and *kar9Δ* cells with normally aligned spindles.
**Source data 4.** Raw and normalized FRAP data of Bfa1-GFP at the SPBs of *spc72Δ* and *kar9Δ* cells with misaligned spindles.

phosphorylated by Cdc5 in *kin4Δ* cells (*Figure 6E*) (*Maekawa et al., 2007*; *Pereira and Schiebel, 2005*). Levels of the hyper-phosphorylated forms of Bfa1 were strongly reduced in *spc72Δ* cells (*Figure 6E*), although Kin4 was not able to phosphorylate Bfa1 in these cells (*Figure 5C*). Thus, not only Kin4 but also Cdc5 requires Spc72 to efficiently phosphorylate Bfa1. Our data altogether suggests that Spc72 serves as a platform that integrates Kin4 and Cdc5 counteracting actions on Bfa1.

## Bfa1-Spc72 interaction enhances Bfa1 phosphorylation by Cdc5 and Kin4

To this point, we show that Spc72 is necessary for Bfa1 phosphorylation by both Cdc5 and Kin4. As Spc72-Bfa1 interaction is disrupted during the SPOC arrest, we reasoned that the disassociation of Bfa1 from Spc72 might be required to prevent the inhibitory phosphorylation of Bfa1 by Cdc5. To test this hypothesis, we constructed cells bearing endogenous *SPC72* C-terminally fused with the GFP-binding protein (GBP). GBP efficiently binds GFP or GFP tagged proteins in human and yeast cells (*Bertazzi et al., 2011*; *Rothbauer et al., 2008*). Spc72-GBP constitutively recruited Bfa1-GFP to both SPBs regardless of the spindle position (*Figure 7A*). Emphasizing the importance of Spc72-Bfa1 disengagement for SPOC, coexistence of Bfa1-GFP and Spc72-GBP in the same cell resulted in SPOC failure (*Figure 7B*). We further analyzed the phosphorylation profile of Bfa1-GFP in these cells. In the presence of Spc72-GBP, Bfa1-GFP accumulated hyper-phosphorylated forms, which disappeared upon depletion of Cdc5 (*Figure 7C*, lanes 3 ,4 and 5). Thus, persistence of Spc72-Bfa1 interaction provokes Cdc5 dependent phosphorylation of Bfa1. Importantly, Kin4 also contributed to the phosphorylation of Bfa1-GFP in Spc72-GBP cells, as deletion of *KIN4* reduced the hyper-phosphorylated forms of Bfa1 (*Figure 7C*, lanes 4 and 6). Our data suggests a model where Bfa1 dissociates from Spc72 to prevent the inhibitory phosphorylation of Bfa1 by Cdc5 (*Figure 7D*).

## Re-alignment of one spindle is sufficient to drive mitotic exit in binucleated *spc72Δ* cells with misaligned spindles

Spindle misalignment is observed in both *SPC72* and *KAR9* knock-out cells. However, in contrast to *kar9Δ* cells, SPOC deficient phenotypes (multiple spindles and/or buds) accumulate with a high frequency (> 40%) in *spc72Δ* cultures (*Figure 8A*) (*Hoepfner et al., 2002*). This was somehow puzzling considering that the SPOC was functional in *spc72Δ* cells (as shown above through the mitotic exit delay upon spindle misalignment) (*Figure 6*). To understand why SPOC deficient phenotypes

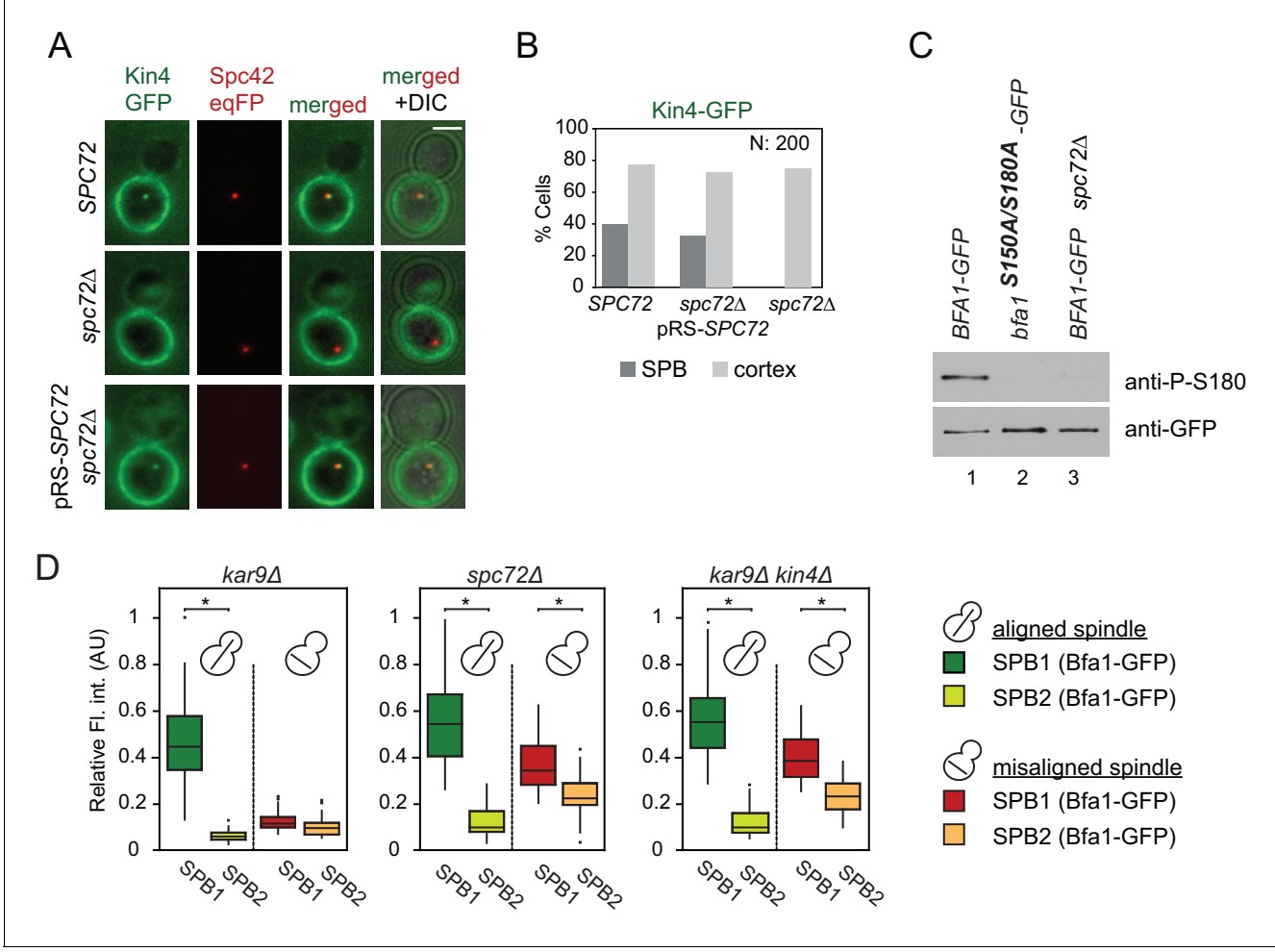

**Figure 5.** Lack of Spc72 interferes with Kin4 localization and functioning at the SPB. (**A**) SPB localization of Kin4-GFP in *SPC72* (pRS-*SPC72 spc72Δ*) and *spc72Δ* cells carrying the SPB marker *SPC42-eqFP*. Cells were arrested in metaphase with nocodazole. (**B**) Quantification of (**A**). (**C**) Immunoblots showing Bfa1 phosphorylation by Kin4 at S180 residue. Bfa1-GFP was immunoprecipitated from indicated strains. Total amount of immunoprecipitated Bfa1-GFP and Bfa1-GFP that is phosphorylated at S180 were detected by anti-GFP and anti-P-S180 antibodies respectively. Bfa1$^{S180A/S150A}$ served as a control for the specificity of anti-P-S180 antibody. A representative blot out of three independent experiments is shown. (**D**) Box and Whisker plots of Bfa1-GFP fluorescence intensity at SPBs in *kar9Δ*, *spc72Δ* and *kar9Δ kin4Δ* cells with correctly and mis-aligned spindles. Within the same cell, the SPB with stronger and weaker Bfa1 fluorescence intensity were classified as SPB1 and SPB2, respectively. The maximum Bfa1-GFP fluorescence intensity of each data set was normalized to 1. The boxes show the lower and upper quartiles, the whiskers show the minimum and maximal values excluding outliers; outliers (shown as dots) were calculated as values greater or lower than 1.5 times the interquartile range; the line inside the box indicates the median. For each box, 33 SPBs were quantified. Asterisks show significant difference according to student's t-test (p<0.01). See the accompanying data file for exact p-values (*Figure 5—source data 1*).

The following source data is available for figure 5:

**Source data 1.** Raw and normalized mean fluorescence intensities of Bfa1-GFP at SPBs of *spc72Δ* and *kar9Δ* cells with normally aligned or misaligned spindles (source data for *Figure 5D*).

accumulated in these populations, we analyzed the behavior of *spc72Δ GFP-TUB1* cells by live cell imaging in greater detail. We monitored the time from anaphase onset until spindle re-alignment. In a *kar9Δ* population most of the cells reoriented their spindle in less than 50 min after anaphase onset. In contrast, spindle mis-orientation persisted for longer than 50 min in the majority of *spc72Δ* cells (*Figure 8B*). The spindles of *spc72Δ* cells that remained arrested in anaphase for an extended period (> 50 min) broke down as a consequence of mitotic exit (*Figure 8C* and *6D*). These cells re-entered the cell cycle to become binucleate cells (*Figure 8D*). This indicated that the SPOC was

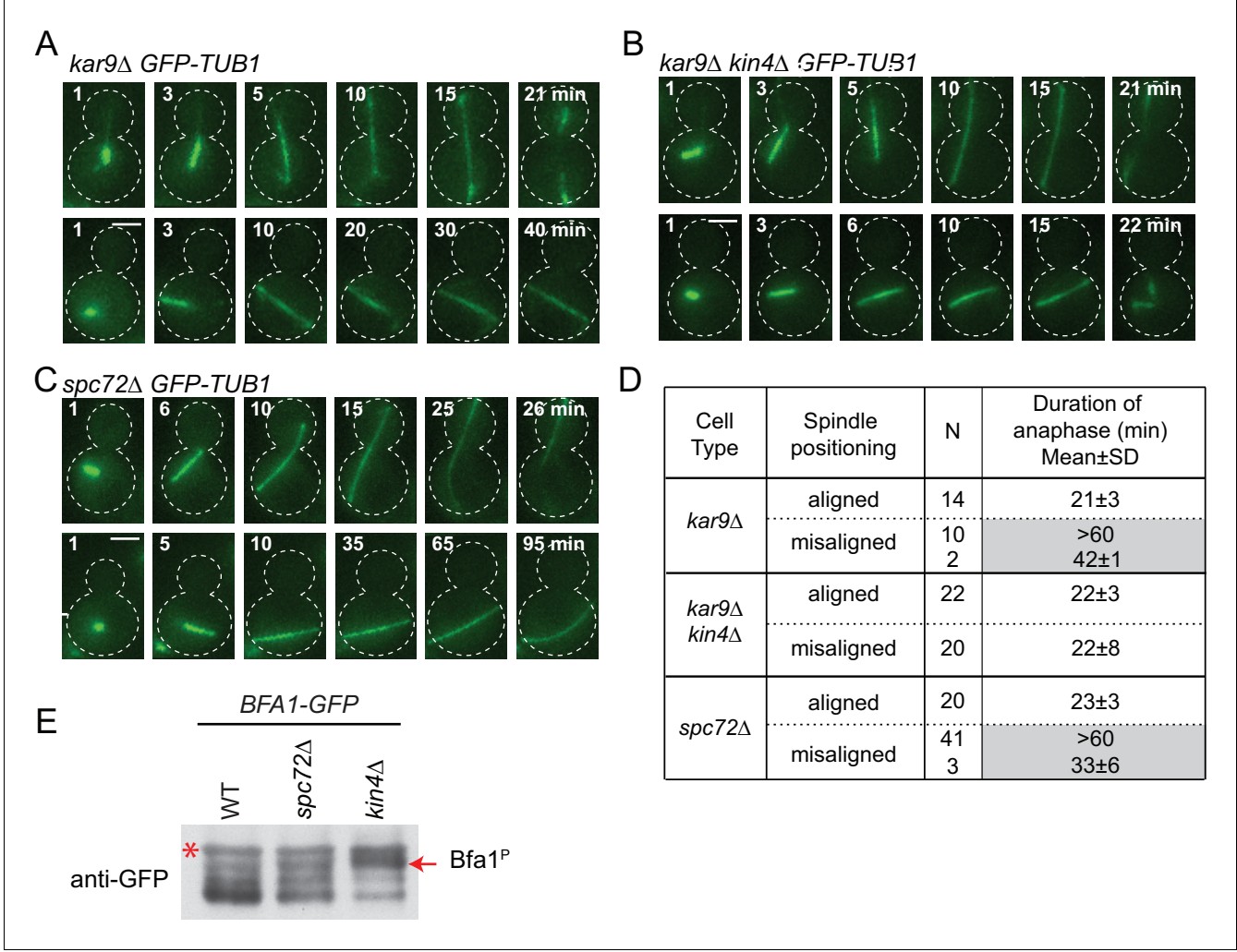

**Figure 6.** *spc72Δ* cells are SPOC proficient (A–C) Representative time-lapse images of *kar9Δ* (A), *kar9Δ kin4Δ* (B) and *spc72Δ* (C) cells carrying *GFP-TUB1*. Time is given in minutes from the start of the inspection. Scale bar 3 μm. (D) Comparison of anaphase duration in *kar9Δ*, *kar9Δ kin4Δ* and *spc72Δ* cells with aligned and misaligned spindles. N: number of cells observed from each category. Mean anaphase duration is given in minutes. SD: Standard deviation. (E) Immunoblot showing Bfa1 mobility shift. Indicated strains were released from G1-block (alpha-factor arrest) in nocodazole containing medium. Samples were collected after 3 hr. Bfa1-GFP was detected in the total cell extracts of indicated strains using anti-GFP antibody. The arrow indicates the hyper-phosphorylated form of Bfa1. Asterisks indicate an unspecific band detected by the anti-GFP antibody. A representative blot out of three independent experiments is shown.

The following figure supplement is available for figure 6:

**Figure supplement 1.** Effect of Bfa1 on the growth of *spc72Δ* cells.

unable to maintain an indefinite cell cycle arrest in this population of cells. Interestingly, in the majority of binucleated cells that entered anaphase (10 out of 15 cells), one spindle remained in the mother cell compartment while the second entered the bud (*Figure 8D*, upper panel). In all such cells, the mother-bud oriented spindle and the misoriented spindle disassembled simultaneously, as the cells exited mitosis (*Figure 8D and E*). This behavior was not due to binucleation per se, because cells harboring two mis-oriented spindles remained arrested in late anaphase with intact anaphase spindles (*Figure 8D*, lower panel, and *Figure 8E*). This indicates that satisfaction of the SPOC by the correctly aligned spindle triggers a dominant signal that silences or bypasses the SPOC throughout the cell. This signal is likely to be a diffusible signal as the breakdown of the correctly aligned and misaligned nuclei occurs simultaneously.

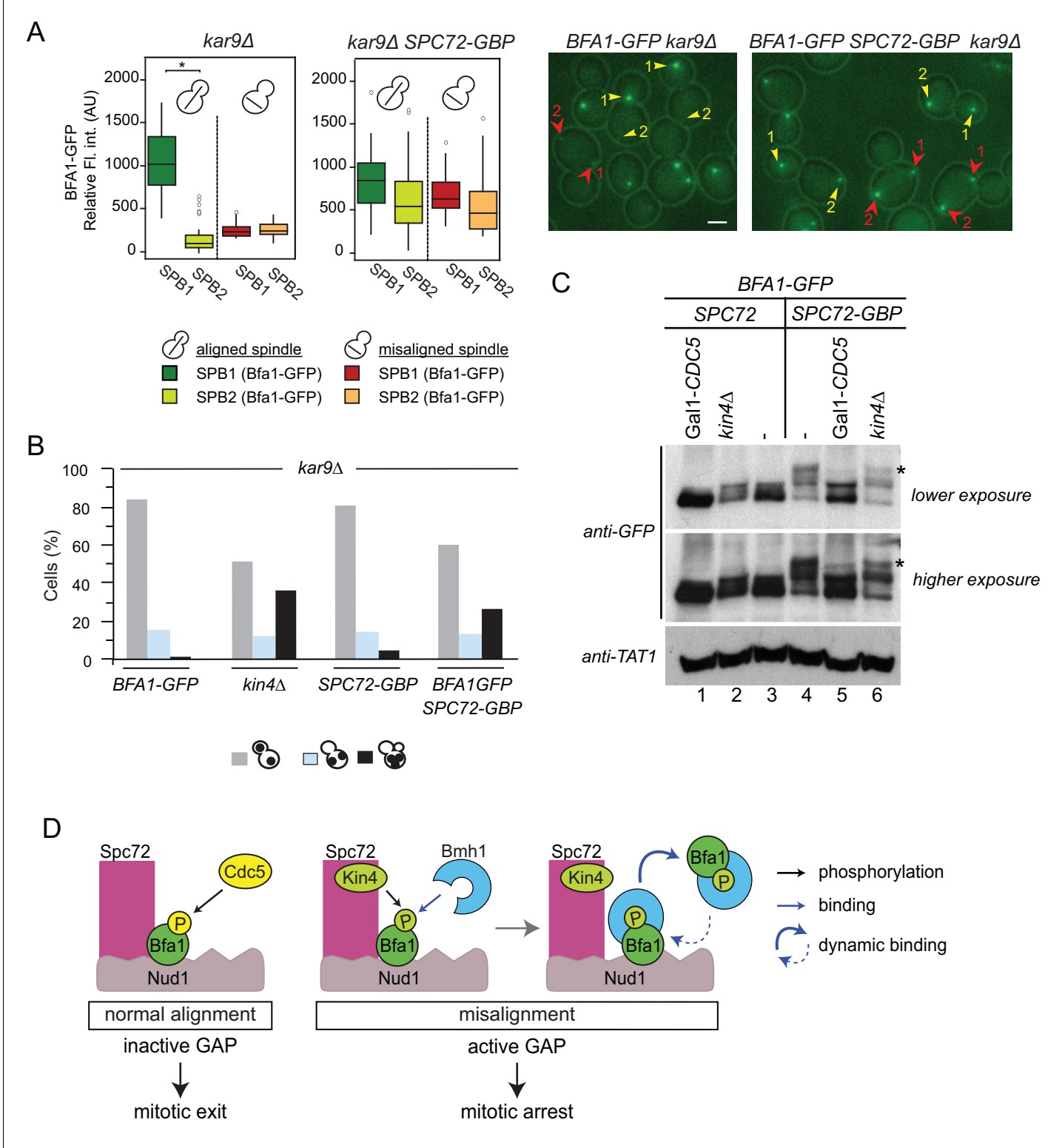

**Figure 7.** Bfa1-Spc72 interaction provokes Cdc5 phosphorylation of Bfa1 (**A**) Still images (right) and the SPB-bound mean fluorescence intensities (left) of Bfa1-GFP in *kar9Δ* and *SPC72-GBP kar9Δ* cells. SPB1 indicates the SPB closer to bud where SPB2 is the SPB closer to the mother cell compartment. Yellow or red arrows point the two SPBs in a cell with a normally aligned or misaligned spindle respectively. The numbers 1 and 2 next to the arrows indicate SPB1 and SPB2 respectively. Scale bar: 3 μm. In the Box-Whisker plots showing Bfa1-GFP mean fluorescence intensities, the boxes show the lower and upper quartiles, the whiskers show the minimum and maximal values excluding outliers; circles represent the outliers calculated as values greater or lower than 1.5 times the interquartile range; the line inside the box indicates the median. Asterisks show significant difference according to

*Figure 7 continued on next page*

*Figure 7 continued*

student's t-test (p<0.001). Sample sizes for *kar9Δ* cells were 49 and 23, whereas sample sizes for *kar9Δ SPC72-GBP* cells were 38 and 14 for each SPB with normal and misaligned spindles respectively. See the accompanying data file for exact p-values (*Figure 7—source data 1*). (B) SPOC integrity of the indicated cell types. Cells were grown at 23°C and shifted to 30°C for 3 hr to induce the accumulation of cells with misaligned spindles (blue bars). Black bars show the percentage of multi-nucleated phenotypes, which indicates SPOC deficiency N: 100 cells per strain. A representative plot out of three biological replicates is shown. (C) Immunoblot showing Bfa1-GFP mobility shift. Indicated strains arrested in G1 using alpha-factor and released in alpha-factor free, nocodazole containing YPDA medium. Samples were collected after 2.5 hr. Gal1-*CDC5* bearing strains were grown and G1-arrested in raffinose and galactose containing medium. Releasing from G1 block in glucose containing medium repressed the Gal1 promoter to maintain Cdc5 depletion. Bfa1-GFP was detected in the total cell extracts of the indicated strains using anti-GFP antibody. Anti-TAT1 antibody was used to detect tubulin as a loading control. Asterisks indicate the hyper-phosphorylated form of Bfa1. A representative blot out of three independent experiments is shown. (D) Model for SPOC activation and Bfa1-SPB remodeling. When the mitotic spindle is correctly aligned, Bfa1 molecules are stably in contact with Nud1 and Spc72, where Cdc5 can phosphorylate and thereby inactivate Bfa1. When the spindle misaligns, Kin4 binds to Spc72 to phosphorylate Bfa1 (*Maekawa et al., 2007*). Kin4 phosphorylated Bfa1 is recognized by Bmh1 (*Caydasi et al., 2014*). Bmh1-bound Bfa1 disconnects from Spc72 but remains associated with Nud1, although dynamically. Cdc5 cannot phosphorylate Bfa1 when Bfa1 is disconnected from Spc72.

The following source data is available for figure 7:

**Source data 1.** Raw and normalized mean fluorescence intensities of Bfa1-GFP at SPBs of *SPC72 kar9Δ* and *SPC72-GBP kar9Δ* cells with normally aligned or misaligned spindles (source data for *Figure 7A*).

---

In conclusion, inefficiency of spindle re-orientation, together with SPOC leakage after prolonged arrest, leads to the accumulation of binucleated cells in *spc72Δ* cells. Binucleated cells, on the other hand, have a higher probability of producing bi- or multinucleated cells in the next round of cell division, as one correctly aligned spindle is sufficient to disable SPOC signaling for both itself and its misaligned neighbor. Altogether these data explain why *spc72Δ* populations accumulate bi- and multiple-nuclei even though they are SPOC proficient.

## Discussion

Spindle pole bodies (SPBs) not only facilitate microtubule nucleation but also provide a scaffolding platform for binding of the mitotic exit network (MEN) and the spindle position checkpoint (SPOC) proteins. One hallmark of SPOC activation is the drastic loss of Bfa1-Bub2 molecules from one of the two SPBs upon spindle mis-positioning. This change is essential for SPOC function and is thought to arise from a decrease in Bfa1-Bub2 residence time at SPBs (*Caydasi and Pereira, 2009*; *Monje-Casas and Amon, 2009*). In this study, we investigated how Bfa1-Bub2 interacts with SPBs by FRET microscopy. Our findings unraveled a novel molecular rearrangement of SPOC proteins at SPBs that is essential for SPOC function. In addition, the analysis of binucleated cells showed that the SPOC is switched off as soon as one spindle enters the bud. This indicates the generation of a dominant and diffusible signal that promotes mitotic exit and SPOC silencing upon checkpoint satisfaction.

### SPOC activation disturbs Bfa1-Spc72 interaction through Kin4/Bmh1 branch of the SPOC

A positive FRET between two fluorophores only occurs when the fluorophores are in close apposition (< 10 nm). This indicates that the fluorophore-fused proteins are in close proximity and likely to physically interact (*Zal and Gascoigne, 2004*). Interactions between SPB core components have been previously mapped by conventional FRET based on acceptor emission measurements (sensitized emission FRET) (*Muller et al., 2005*). In comparison to sensitized emission FRET, acceptor photobleaching FRET yielded more robust measurements for Bfa1-Bub2 SPB interactions. Our FRET data show that the C-terminal domain of Bfa1 resides in close proximity to the C-termini of the γ-tubulin receptor protein Spc72 and the MEN scaffold protein Nud1 at the SPB outer plaque. Direct physical interaction of Nud1 and Bfa1 was already shown (*Gruneberg et al., 2000*). In vitro binding assays performed here established that Bfa1 also associates with Spc72. Thus, FRET interactions between Bfa1-Nud1 and Bfa1-Spc72 most likely reflect physical interactions at SPBs in vivo. C-terminally tagged Bfa1 also yielded positive FRET with N-terminally tagged Bub2 at SPBs, which is in agreement with the fact that Bfa1 and Bub2 bind to SPBs as a bipartite complex (*Caydasi and Pereira, 2009*; *Pereira et al., 2000*). We could not detect FRET between the N-terminally tagged Bub2

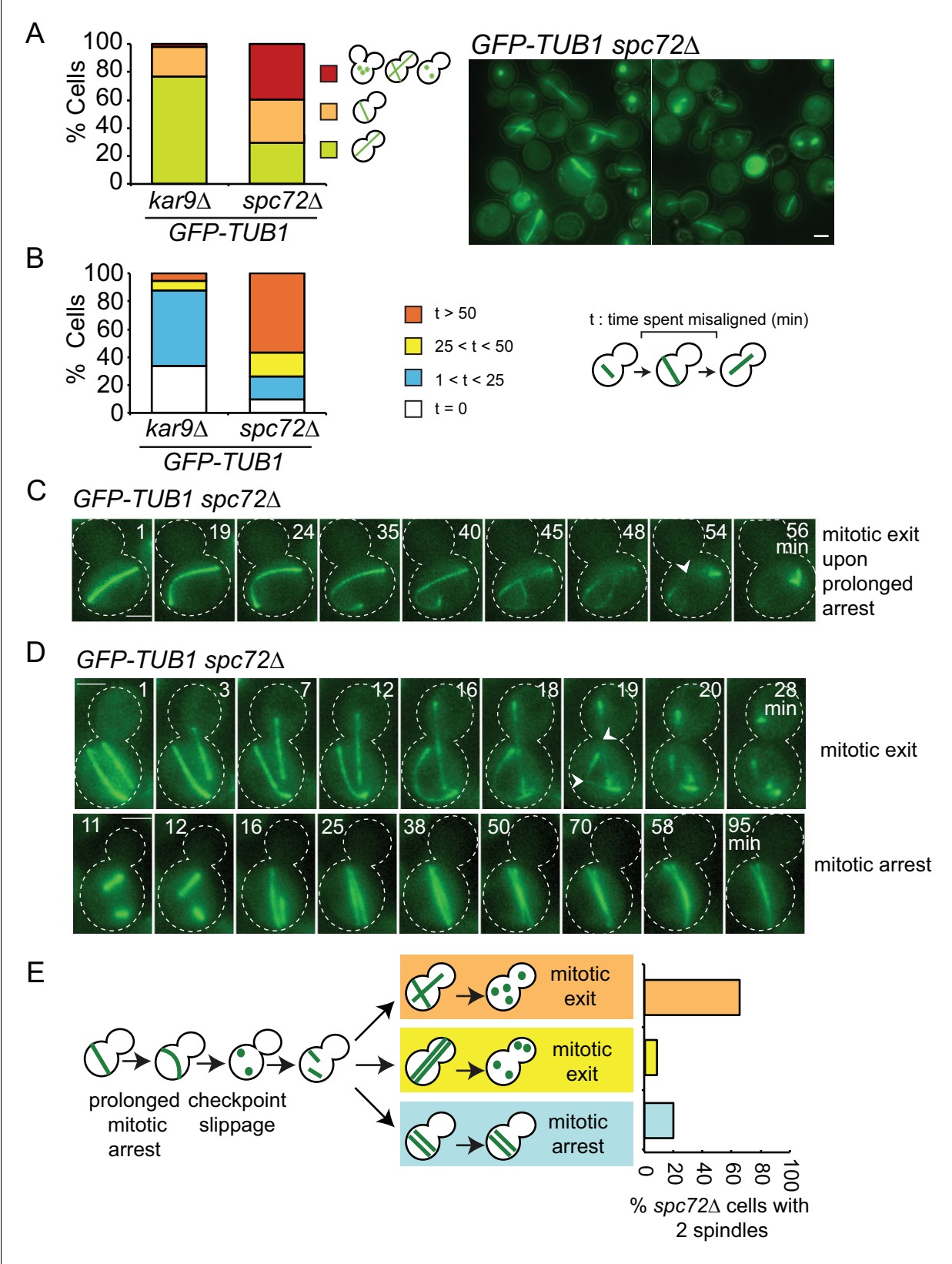

**Figure 8.** SPOC slippage in *spc72Δ* cells (**A**) Population analysis of *GFP-TUB1 kar9Δ* and *GFP-TUB1 spc72Δ* cells. Percentage of cells with single or multiple spindles is indicated in the left panel. Representative images of *GFP-TUB1 spc72Δ* cells are shown in the right panel. Fixed samples were used
*Figure 8 continued on next page*

*Figure 8 continued*

for analysis (N>100). Note that the presence of more than one *GFP-TUB1* signal in cells is an indication of multiple nuclei. (B) Duration of spindle misalignment in *kar9Δ GFP-TUB1* and *spc72Δ GFP-TUB1* cells. Time-lapse series were used for analysis (N: 99 for *kar9Δ*, N: 54 for *spc72Δ*). The time elapsed between the anaphase onset and spindle orientation in the mother to daughter direction was recorded as the time spent misaligned (t). T=0 indicates that no spindle misalignment was observed in anaphase. (C) Representative images from time-lapse series of a *GFP-TUB1 spc72Δ* cell with a mis-aligned spindle that exited mitosis after prolonged arrest. Time point one (min) is the first time point after the start of the time-lapse. Please note that the cell depicted here had a misaligned anaphase spindle already at time point one. Spindle breakdown (t=54) is an indication of mitotic exit. (D) Representative images from time-lapse series of *GFP-TUB1 spc72Δ* cells with two spindles. In the upper panel, one spindle stays misaligned while the other one re-aligns in the mother-daughter direction. In this cell, both spindles broke at t=19, indicating mitotic exit. In the lower panel, both spindles stay misaligned during the course of the experiment. Mitotic exit was not observed in this cell. (E) Diagram depicting the fate of an *spc72Δ* cell with a misaligned spindle. Upon prolonged spindle misalignment, mitotic exit occurs (checkpoint slippage). The binucleated cell enters a second cell cycle with two nuclei, each forming a spindle. The two spindles could both misalign (lower panel) or correctly align (middle panel). These two events are rare (graph in the right panel). Alternatively, one of the two spindles could stay misaligned while the other is correctly aligned (upper panel). This occurs more frequently (graph in the right panel). The frequencies of each three scenarios plotted on the right were calculated based on time-lapse analysis of *GFP-TUB1 spc72Δ* cells (N: 15 cells). Arrowheads indicate the point of spindle breakage. Scale bar: 3 μm. Data represented is a collection of two independent experiments.

and Nud1 (or Spc72). However we cannot exclude the possibility that Bub2 may interact with Nud1 or Spc72 by other means.

We observed loss of Bfa1-Spc72 (but not Bfa1-Nud1) FRET upon SPOC activation. The loss of FRET signal between Bfa1 and Spc72 might indicate the physical separation of the two proteins or only the separation of the fluorophores coupled to Bfa1 and Spc72 C-termini. Forced binding of Bfa1 to Spc72 resulted in loss of SPOC activity, suggesting that dissociation of the two proteins is important for checkpoint functioning. Our data show that Kin4 kinase and the 14-3-3 protein Bmh1 are involved in this process. Loss of Bfa1-Spc72 FRET interaction during spindle misalignment was only observed in *KIN4* but not in *kin4Δ* cells. Furthermore, overexpression of *KIN4* abolished the FRET signal between Bfa1 and Spc72 in cells with normal aligned anaphase spindles in a Bmh1-dependent manner. We propose that the remodeling of Bfa1-SPB binding is triggered by Kin4 phosphorylation of Bfa1. As a direct consequence of this phosphorylation, Bmh1 binds to Bfa1 and breaks Bfa1-Spc72 proximity (*Figure 7D*) (*Caydasi et al., 2014*). The reorganization of Bfa1-SPB binding may result from a possible Bfa1 conformational change triggered by recruitment of Bmh1 to Kin4 phosphorylated Bfa1. Such regulation may resemble the conformational change that Mad2 undergoes upon activation of the spindle assembly checkpoint (SAC) (*Luo et al., 2002*; *2004*; *Mapelli et al., 2007*; *Mapelli and Musacchio, 2007*; *Sironi et al., 2002*; *Yang et al., 2007*).

## The γ-tubulin complex receptor Spc72 coordinates activation and inhibition of Bfa1-Bub2

Spc72 has been proposed to constitute a part of the SPOC mechanism that senses cytoplasmic microtubule failure and recruits Kin4 to SPBs in response to this defect (*Maekawa et al., 2007*). Surprisingly, *spc72Δ* cells were SPOC proficient even though Kin4 neither localized at SPBs nor phosphorylated Bfa1. However, Cdc5-dependent hyper-phosphorylation of Bfa1, which inactivates the Bfa1-Bub2 GAP complex (*Geymonat et al., 2003*; *Hu et al., 2001*), was greatly reduced in the absence of *SPC72*. The loss of Bfa1 phosphorylation by Cdc5 in the absence of Spc72 probably explains why the SPOC is still functional in these cells despite the lack of Kin4 phosphorylation of Bfa1. Given that Cdc5 interacts with Spc72 and phosphorylates Bfa1 at the SPB (*Archambault and Glover, 2008*; *Maekawa et al., 2007*; *Park et al., 2004*; *Snead et al., 2007*), it is tempting to speculate that Cdc5 only phosphorylates Bfa1 when in complex with Spc72. In this case, disruption of the Bfa1-Spc72 interaction in cells with mis-oriented spindles might be a way to insulate the GAP complex from being inhibited by Cdc5 (*Figure 7D*). We tested this model using *BFA1-GFP SPC72-GBP* cells where Spc72-Bfa1 interaction is maintained even during spindle misalignment. In support of our model, Bfa1-Spc72 tethering caused increased phosphorylation of Bfa1 by Cdc5 and hence SPOC failure. Interestingly, phosphorylation of Bfa1 by Kin4 was also observed in these cells. This indicates that Kin4 can only inhibit Bfa1 phosphorylation by Cdc5 if Bfa1 dissociates from Spc72. This is in line with the observation that Bfa1 phosphorylated by Kin4 is still a substrate of Cdc5 in vitro (*Maekawa et al., 2007*). We therefore propose that Spc72 acts as a scaffold protein that

coordinates the regulation of the checkpoint effector Bfa1 by both Kin4 and Cdc5 kinases in cells with mis-aligned spindle.

## Contribution of Nud1 and Spc72 for Bfa1 SPB binding behavior

Biochemical and FRET data are consistent with the binding of Bfa1 to Nud1 and Spc72. This raises the question of how Bfa1 associates with these proteins at the SPB. Bfa1 could bind separately or simultaneously to Nud1 and Spc72. The analysis of *spc72Δ* cells exclude the possibility of a stable Bfa1-Spc72 and a dynamic Bfa1-Nud1 pool co-existing at SPBs since Bfa1 still associated with SPBs via Nud1 in a stable manner in *spc72Δ* cells with correctly aligned spindles. Interestingly, in *spc72Δ* cells with mis-aligned spindles, Bfa1 became dynamic, yet not as much as in wild type cells but similar to *kin4Δ* cells. This observation is important two-fold: First, it indicates that the Bfa1 binding site of Nud1 functions efficiently without Spc72. Second, it suggests that the main function of Spc72 is not in Bfa1 binding to the SPB but instead in the SPB recruitment of Kin4, which then regulates Bfa1 SPB dynamics through phosphorylation of Bfa1 as discussed above.

Simultaneous binding of Bfa1 to Spc72 and Nud1 is a possibility considering the SPB structure, where Nud1 and Spc72 interact through their C-termini with each other and with Bfa1 (*Gruneberg et al., 2000*). If this is the case, the binding affinity of Bfa1 to Nud1 must be higher than to Spc72. This would explain why the loss of Spc72 does not significantly influence Bfa1 protein levels and binding dynamics at SPBs. However, alternative scenarios might also explain why Bfa1 has the same SPB binding behavior in wild type and *spc72Δ* cells with properly aligned spindles. For example, an adaptation mechanism that activates an additional SPB binding site for Bfa1 at SPBs may compensate for the loss of Spc72 or Bfa1 might not directly bind to Spc72 in vivo. We consider the latter possibility as unlikely because of the positive Bfa1-Spc72 FRET signal at the SPB, demonstrating the very close neighborhood of both proteins at this organelle. Moreover, we observed changes in Bfa1-Spc72 FRET signal that are consistent with previous dynamic measurements of Bfa1 in response to spindle alignment defects or *KIN4* overexpression. In any case, further biochemical and biophysical studies will be necessary to evaluate the affinity of Bfa1 towards Nud1 and Spc72, and to establish whether the same Bfa1 molecule can bind simultaneously to Nud1 and Spc72.

## Asymmetric association of Bfa1 with SPBs in cells with correctly aligned spindles

The Bfa1-Bub2 complex is recruited preferentially to the dSPB (asymmetric binding) in cells progressing normally through the cell cycle. How this asymmetry is established is still unclear. SPB duplication generates a newly formed SPB next to the old one (*Byers and Goetsch, 1974*; *Winey et al., 1991*). Under normal growth conditions, the old SPB moves to the daughter cell body (*Juanes et al., 2013*; *Pereira et al., 2001*). Recently, it was shown that the old SPB binds more Spc72 molecules then the new one (*Juanes et al., 2013*), raising the possibility that Spc72 could be the determinant of Bfa1 SPB binding asymmetry. However, our data indicate that Bfa1 asymmetry is maintained even when Spc72 is absent through stable binding to Nud1. In addition, randomization of SPB inheritance (through transient microtubule depolymerization) did not alter the asymmetric association of Bfa1 with the dSPB (*Juanes et al., 2013*; *Pereira et al., 2001*). Furthermore, our FRET analyses show that, in cells with normal aligned spindles, Bfa1 is still in close proximity to Spc72 and Nud1 at both SPBs. We thus consider it as unlikely that the asymmetric SPB binding of Bfa1-Bub2 arises from association of Bfa1 with different receptors at mother and daughter SPBs.

The asymmetry of Bfa1 at SPBs could arise from differential phospho-regulation at the poles. This regulation could be at the level of Bfa1, Nud1 or Bub2, which is also required for Bfa1 SPB asymmetry and subjected to phospho-regulation (*Hu and Elledge, 2002*; *Maekawa et al., 2007*; *Park et al., 2008*; *Rock et al., 2013*). Phosphorylation could either stabilize Bfa1-Nud1 interaction at the dSPB or destabilize it at mSPB. Cdc5 has been previously suggested to preferentially target Bfa1 to the dSPB (*Kim et al., 2012*). However, the fact that Bfa1 asymmetry was not disturbed in Cdc5-depleted cells challenges this view (*Caydasi and Pereira, 2009*). Existing data indicated that Kin4 does not contribute to the reduced levels of Bfa1 at the mSPB (*Caydasi and Pereira, 2009*). In addition, our data now shows that Bfa1 asymmetry is not disturbed in *spc72Δ* cells, where Cdc5 and Kin4 do not phosphorylate Bfa1. These observations suggest that kinases other than Kin4 and Cdc5 are involved in Bfa1 asymmetry, if the regulation is at the level of phosphorylation.

It has been proposed that Bfa1-Bub2 asymmetry is established through cytoplasmic microtubule-cortex interactions (*Caydasi and Pereira, 2009*; *Monje-Casas and Amon, 2009*; *Pereira et al., 2001*) or cell polarity determinants (*Monje-Casas and Amon, 2009*). Bfa1 asymmetry was maintained in *spc72Δ* cells, which have very short and unstable cytoplasmic microtubules (*Hoepfner et al., 2002*). This implies that cells do not require an intact cytoplasmic microtubule cytoskeleton to establish and/or maintain Bfa1-Bub2 asymmetry. How cell polarity determinants control Bfa1 asymmetry is unclear. Daughter cell associated factors could stabilize the Bfa1-Nud1 interaction at the dSPB, for example by influencing the post-translational regulation of Bfa1-Bub2 or Nud1.

## Local SPOC satisfaction triggers a global dominant signal promoting mitotic exit

Our data show that SPOC is not an everlasting checkpoint. Slippage from SPOC can occur after prolonged mitotic arrest. In *kar9Δ* cells, this is a relatively rare event because the spindle is able to realign quickly in the majority of the cells due to the presence of the alternative dynein-dependent spindle alignment pathway (*Eshel et al., 1993*; *Li et al., 1993*). This situation is however different in *spc72Δ* cells. Lack of functional cytoplasmic microtubules impairs spindle realignment in the majority of the *spc72Δ* cells. As a consequence of this long cell cycle arrest, we frequently observed SPOC slippage and accumulation of multi-nucleated cells. Our data is consistent with a previous report that analyzed the cell cycle progression of binucleated yeast cells lacking cytoplasmic microtubules (*Sullivan and Huffaker, 1992*). By following *spc72Δ* cells that underwent SPOC slippage and became multinucleated, we observed a novel phenomenon in yeast mitotic exit: In cells with two misaligned anaphase spindles, realignment of one spindle was sufficient to trigger mitotic exit in both misaligned and correctly aligned nuclei. A similar phenomenon was described in binucleated budding yeast cells obtained by other means (*Falk et al., 2016*). This observation indicates that a dominant mitotic exit signal is generated after spindle entry into the bud. This is likely a diffusible signal as it almost simultaneously affects the other nuclei that still reside in the mother cell compartment with a misaligned spindle.

The passage of one nucleus into the bud may trigger mitotic exit via several possible SPOC bypassing and/or silencing mechanisms. Daughter cell specific factors, such as the putative Tem1 guanine nucleotide exchange factor Lte1 (*Adames et al., 2001*; *Bardin et al., 2000*; *Keng et al., 1994*; *Pereira et al., 2000*; *Shirayama et al., 1994*) could trigger the MEN to initiate mitotic exit in the bud. Active GTP-bound Tem1 would then fully activate the phosphatase Cdc14 that in turn counteracts and decreases the activity of mitotic cyclin-dependent kinase (Cdk) to promote mitotic exit and cytokinesis (*Bardin and Amon, 2001*; *Meitinger et al., 2012*). In this case, Tem1 activation in the daughter cell followed by mitotic Cdk1 down-regulation would bypass the SPOC in the mother cell compartment (SPOC bypassing mechanism). Interestingly, Lte1 is also a Kin4 inhibitor (*Bertazzi et al., 2011*; *Falk et al., 2011*), implying that bud specific factors may promote MEN activation through inhibition of SPOC components (SPOC silencing). SPOC silencing can also occur at the level of Bfa1-Bub2. Cdc5 was proposed to inhibit Bfa1-Bub2 GAP activity in cells with normal aligned spindle (*Geymonat et al., 2002*; *2003*; *Hu and Elledge, 2002*). However, it is likely that factors other than Cdc5 inactivate Bfa1-Bub2 in the daughter cell compartment. Two observations support this notion. We found that Bfa1-Bub2 was not efficiently phosphorylated by Cdc5 in *spc72Δ* cells, yet these cells did not have a delayed mitotic exit when their mitotic spindle was correctly aligned. In addition, a mutant form of Bfa1 (Bfa1-11A), which cannot be phosphorylated by Cdc5, does not cause a delayed mitotic exit (*Hu et al., 2001*). Mathematical modeling also predicted the existence of an additional Bfa1-Bub2 inhibitory mechanism (*Caydasi et al., 2012*). Indeed, the Cdc42 effector protein Gic1 was shown to directly bind to Bub2 and inhibit its interaction with Tem1 (*Hofken and Schiebel, 2004*). Our study thus re-invigorates the concept that cell polarity associated factors play a critical role in SPOC silencing and mitotic exit (*Hofken and Schiebel, 2002*; *2004*; *Monje-Casas and Amon, 2009*). The molecular characterization of these factors, which were undermined in the last decade, will be critical to shed light onto the mechanisms controlling mitotic exit and/or SPOC silencing.

Of importance, our data indicate that the mitotic exit stop-signal generated by the SPOC at the misaligned spindle cannot sustain the mitotic arrest if the checkpoint is satisfied in the neighboring spindle. Analogously, in mammalian cells with two mitotic spindles, the stop-anaphase signal

generated by the SAC at one of the two spindles was not sufficient to sustain the checkpoint arrest if the checkpoint was satisfied at the neighboring spindle (*Rieder et al., 1997*). Similar observations were made for G2-M checkpoint in binucleated plant cells (*del Campo et al., 1997*; *Gimenez-Abian et al., 2001*). Therefore, local satisfaction of checkpoints seems to be sufficient for global checkpoint silencing in binucleated cells. Interestingly, cytokinesis defects often generate binucleation, which is associated with aneuploidy and tumor formation. We postulate that misbalanced checkpoint integrity in binucleated cells might be a driver for genome instability and cancer development.

## Materials and methods

### Strains and growth conditions

All yeast strains and plasmids used in this study are listed in the *Supplementary file 1* (*Baudin-Baillieu et al., 1997*; *Caydasi et al., 2010*; *2014*; *Gruneberg et al., 2000*; *Khmelinskii et al., 2007*; *Knop and Schiebel, 1998*; *Maekawa et al., 2007*; *Pereira et al., 2001*; *Sikorski and Hieter, 1989*; *Surana et al., 1993*). Yeast strains of ESM356-1 or YPH499 background are derivatives of S288C, strains of BMA64 background are isogenic with W303. Basic yeast methods and growth media were previously described (*Sherman, 1991*).

For live-cell imaging yeast strains were grown to log-phase in filter sterilized synthetic complete media (SC) at 30°C. Plasmid-bearing strains were grown in SC media lacking the selective amino acid. To cure yeast strains of *URA3*-based plasmids, cells were grown on 5-fluoroorotic acid (5FOA, 1 mg/ml) containing agar plates. Expression from Gal1 promoter was induced by adding galactose (2% final concentration) into log-phase yeast cultures grown in 3% raffinose-containing media. For repression of the Gal1 promoter (i.e. Gal1-*CDC20* and Gal1-*CDC5* depletion), cells grown in raffinose and galactose containing medium were transferred into glucose containing medium. For depletion of Cdc5, cells were arrested in G1 using alpha factor prior to their transfer in glucose containing medium. G1-phase arrest was achieved by adding 10 µg/ml synthetic alpha-factor (Sigma, St. Louis, MO) to the logarithmically growing (log-phase) culture and further incubation of the culture until >90% of cells formed mating projections. To induce microtubule depolymerization, 15 µg/ml nocodazole (Sigma) was added to the log-phase yeast cultures grown in yeast/peptone/dextrose medium with 0.1 mg/l adenine (YPDA) at 30°C until the mid-log phase ($10^7$ cells/ml). For protein immunoprecipitation, cell pellets were collected after 2.5 hr of incubation in nocodazole containing medium. *kar9Δ* cells were grown at 23°C until and shifted to 30°C for 2–4 hr prior the experiment.

### Construction of strains and fluorescent fusions

PCR-based strategy was used for gene deletions, tagging of yeast proteins with fluorophores and construction of Gal1-*KIN4* strains (*Janke et al., 2004*; *Knop et al., 1999*). Plasmids to generate FRET fusions encoded monomeric versions of YFP and mTUR with A206K mutation (*Zacharias et al., 2002*). *mCherry-TUB1*, Gal1-*clb2ΔDB*, *GFP-TUB1* were integrated in the yeast genome by homologous recombination using yeast integration plasmids (*Caydasi et al., 2010*). N terminal tagging of proteins with Gal1-*mTUR or* Gal1-*EYFP* using the plasmids pYG2 and pYG3 resulted in only moderate expression of the corresponding protein, as the levels of Gal1-*mTUR or* Gal1-*EYFP* tagged proteins were only slightly increased in comparison to endogenous levels of the corresponding proteins (data not shown).

### Fluorescence microscopy

For FRAP, time lapse microscopy and FRET analysis, cells grown in filter sterilized growth media were imaged after attachment of the cells on glass-bottom dishes (MatTek, Ashland, MA) or on 96-well glass-bottom microwell plates (Matrical Bioscience MGB096-1-2-LG, Spokane, WA) coated with 6% concanavalin A-Type IV (Sigma). Cells were fixed with 70% Ethanol prior to nuclear staining using 1 µg/ml 4′,6-diamidino-2-phenylindole (DAPI, Sigma) in PBS.

FRAP experiments, time lapse analysis of *GFP-TUB1* expressing cells and still images of Kin4-GFP and Bfa1-GFP were performed using DeltaVision RT system with softWoRx software (Applied Precision, Issaquah, WA) equipped with a camera (Photometrics CoolSnap HQ; Roper

Scientific, Tucson, AZ). Images were acquired with 2 x 2 binning on a 100x UPlanSAPO objective with a 1.4 NA (Olympus, Tokyo, Japan) and a Mercury arc light source. For time-lapse movies, 12 z-stacks of 0.3 μm optical section spacing were taken at each time point. Movies were taken for 90 min with 1 min time interval. The z-stacks were sum-projected using SoftWoRx software before image analysis. Images for other microscopy experiments were acquired using a Zeiss Axiophot microscope equipped with a 100x NA 1.45 Plan-Fluor oil immersion objective (Zeiss, Jena, Germany), Cascade 1K CCD camera (Photometrics, Tucson, AZ) and MetaMorph software (Universal Imaging Corp., Chesterfield, PA).

Fluorescence intensity measurements, FRAP experiments and FRAP data analysis was as described in *Caydasi and Pereira (2009)*. The protocol for FRET analysis is explained in the following section thoroughly.

## Acceptor photobleaching FRET analysis

Strains for FRET were grown in Synthetic Defined Low Fluorescence media prepared with 6.9 g/l yeast nitrogen base lacking folic acid and riboflavin (SD LoFluo, Formedium, Norfolk, UK) supplemented with all standard amino acids (*Sherman, 1991*) and 2% glucose or 3% raffinose and 2% galactose.

FRET microscopy was performed using a wide-field microscope (Olympus) equipped with a 150 W Mercury-Xenon arc burner. Images were acquired with a 60x UPLFLN air objective (N/A=0.9; Olympus) and on Hamamatsu C9100 EM-CCD camera (Hamamatsu, Japan). EM gain was set to 165 in all experiments. Emission from mTUR was recorded from one plane after focusing at the DIC channel. Fluorescence of mTUR was detected from 417 nm to 451 nm before and after photobleaching of the EYFP. EYFP photobleaching was induced by a 4 s laser pulse of 100 mW 515 nm solid-state laser (Cobolt, Sweden). Images were acquired in the order as presented in *Figure 1—figure supplement 1B*.

Images were analyzed using ImageJ software by measuring mean fluorescence intensities of mTUR signals around SPBs from the part of the image where the EYFP signal was bleached as depicted in *Figure 1—figure supplement 1B*. Fluorescence intensities were corrected for the background and the FRET efficiency was calculated as the percentage increase in corrected mTUR signal after photobleaching of EYFP (*Kentner and Sourjik, 2009*). FRET efficiency values were finally corrected for the median value of the donor-only control (corresponding reference strain with mTUR fusion only, *Figure 1—figure supplement 1D*). To determine if FRET efficiency reflects energy transfer, FRET results for the pair and corresponding negative control were compared using the two-tailed Student t-test. p<0.01 was considered statistically significant.

## Protein methods

Yeast protein extracts and immunoblotting were performed as described in *Janke et al., 2004*. Antibodies were rabbit anti-GFP antibody, goat anti-GST (GE Healthcare, Waukesha, WI), mouse anti-His antibody (GE Healthcare), rabbit anti-P-S180 (*Maekawa et al., 2007*), mouse anti-TAT1 (Sigma) and rabbit anti-Kin4 (lab collection, see the next section). The anti-GFP antibody used in *Figure 7* was a gift from E. Schiebel (ZMBH, University of Heidelberg, Germany) whereas the anti-GFP antibody used in other experiments was a gift from J. Lechner (BZH, University of Heidelberg, Germany). Secondary antibodies were goat anti-mouse, goat anti-rabbit or rabbit anti-goat IgGs coupled to horseradish peroxidase (Jackson ImmunoResearch Laboratories Inc, West Grove, PA).

## Generation of anti-Kin4 polyclonal antibodies

Rabbit polyclonal antibodies were generated against bacterially purified 6His-Kin4 lacking the last 50 amino acids (PSL GmbH, Heidelberg, Germany). Antibodies were purified from preabsorbed serum by affinity purification using the immobilized antigen.

## Recombinant protein purifications and in vitro binding assays

MBP-Bfa1 and GST-Spc72 were purified from *Escherichia coli* as described previously (*Geymonat et al., 2009*; *Maekawa et al., 2007*). 6His-Spc72$^{231-622}$ (Spc72-C) and 6His-Mlc1 were purified from *E. coli (BL21)* according to the manufacturer instructions (EMD Biosciences, San Diego, CA). In vitro binding assays were performed as following: Purified GST-Spc72, 6His-Spc72-C, 6His-

Mlc1 were incubated with bacterially purified MBP-Bfa1 or MBP bound amylose beads (New England Biolabs, Ipswich, MA) in PBS at 4°C for 45 min. MBP-Bfa1 and MBP–bound beads were then washed four times with PBS containing 0.75% Nonidet P-40, and samples were heated at 65°C for 15 min in HU-DTT buffer (200 mM Tris-HCl, pH 6.8, 8 M urea, 5% SDS, 0.1 mM EDTA, 0.005% bromophenol blue, and 15 mg/ml DTT) before loading on SDS–PAGE gels.

## Immunoprecipitation experiments

Immuprecipitation of Bfa1-GFP was done using a cell pellet derived from a 200 ml cell culture ($2 \times 10^7$ cells/ml) treated with nocodazole until >95% of the cells were arrested in metaphase. Cells were lysed using acid-washed glass beads in a FastPrep FP120 Cell Disturber (MP Biomedicals, Irvine, CA). Lysis buffer contained 50 mM Tris-HCl buffer (pH 7.5), 150 mM NaCl, 5% glycerol, 350 µg/ml benzamidine, 100 mM β-glycerophosphate, 50 mM NaF, 5 mM NaVO$_3$ and complete EDTA-free protease inhibitor cocktail (Roche, Basel, Switzerland). Lysates were incubated with Triton X-100 (1% final concentration) for 15 min at 4°C. Total extracts were cleared by centrifugation at 10,000 g for 20 min at 4°C. Bfa1-GFP was precipitated from total cell extracts using GBP conjugated with Sepharose 4B (*Lin et al., 2014*) (gift from E. Schiebel). Phosphorylation of the immunoprecipitated Bfa1-GFP at S180 residue was detected by immunoblotting using anti-P-S180 antibodies. On the same immunoblot, the level of immunoprecipitated Bfa1-GFP was detected using anti-GFP antibodies.

## Acknowledgement

We thank Angelika Amon for communicating results prior to publication. We acknowledge Elmar Schiebel, Johannes Lechner and Michael Knop for sharing reagents and equipment. We are grateful to Elmar Schiebel, Ian Hagan and our lab members for critical comments on the manuscript. This work is supported by the DFG grants PE1883-1/2 granted to GP and DKFZ-ZMBH Alliance bridging project granted to GP and VS; AKC is supported by the DFG (PE1883-1/2); GP is supported by the Heisenberg Program of the DFG (PE1883-2); YG was supported by a fellowship funded by the DKFZ-ZMBH Alliance bridging project; YG was a member of the Hartmut Hoffmann-Berling International Graduate School of Molecular and Cellular Biology (HBIGS) of the University of Heidelberg.

## Additional information

### Funding

| Funder | Grant reference number | Author |
| --- | --- | --- |
| Deutsche Forschungsgemeinschaft | PE1883-1/2 | Ayse Koca Caydasi Gislene Pereira |
| Deutsche Forschungsgemeinschaft | PE1883-2 | Gislene Pereira |

The funders had no role in study design, data collection and interpretation, or the decision to submit the work for publication.

### Author contributions

YG, AKC, Conception and design, Acquisition of data, Analysis and interpretation of data, Drafting or revising the article; GM, Acquisition of data, Analysis and interpretation of data, Drafting or revising the article; VS, GP, Conception and design, Analysis and interpretation of data, Drafting or revising the article

### Author ORCIDs

Gislene Pereira, http://orcid.org/0000-0002-6519-4737

## Additional files

**Supplementary files**
• Supplementary file 1. Table of yeast strains and plasmids. Descriptions of the yeast strains and plasmids used in this study are listed in this file.

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
