## [Decision Letter]

Thank you for submitting your work entitled "A FRET-based study reveals site-specific regulation of spindle position checkpoint proteins at centrosomes" for consideration by *eLife*. Your article has been favorably evaluated by Richard Losick as the Senior editor and three reviewers, one of whom, Yukiko Yamashita, is a member of our Board of Reviewing Editors.

The reviewers have discussed the reviews with one another and the Reviewing Editor has drafted this decision to help you prepare a revised submission.

Summarized review comments:

This is an interesting and important study that addresses the mechanism of SPOC (spindle orientation checkpoint) that ensures proper nuclear segregation prior to mitotic exit. SPOC is a critical checkpoint mechanism that regulates mitotic exit (i.e. upstream of mitotic exit network, MEN) depending on the state of spindle positioning within the cell to ensure correct segregation of two nuclei into two daughter cells. An elegant series of FRET experiments are used to assess the association of Bfa1 with the SPB components Nud1 and Spc72. The data show that Bfa1 simultaneously associates with Spc72 and Nud1 but that it is the association with Spc72 that alters in the SPOC dependent manner.

The reviewers agreed that this study provides a significant progress toward the understanding of molecular mechanism for SPOC. The novel role of Spc72 as a scaffold for Cdc5 and Kin4 is the key point of this study.

In this regard, the major concern was that FRET data are still correlational because the functional significance of Spc72 interactions (measured by FRET) with SPOC components is not defined. So, this correlational nature has to be mentioned in the text.

In the most rigorous manner, one could 1) map the interaction domains of Spc72 that recruits Cdc5 and Kin4, then 2) generate separation of function alleles to define the phenotypic outcome, which will provide a fine model on how Spc72 operates in SPOC signaling, and 3) use FKBP-FRB system to test the model by manipulating Spc72's interaction with SPOC components.

However, the reviewers also agreed that the above points 1 and 2 may be beyond the scope of the current work, and that the existing data presented in this manuscript are sufficient to formulate a working model (as presented in Figure 6). Therefore, the reviewers agreed that the authors should describe their model of how the differential change in Bfa1-Spc72 and Bfa1-Nud1 FRET relates to the ability of Cdc5 and Kin4 to phosphorylate Bfa1, and test the model by additional, relatively simple experiment(s) along the line of above point 3).

[Editors' note: further revisions were requested prior to acceptance, as described below.]

Thank you for submitting your article "A FRET-based study reveals site-specific regulation of spindle position checkpoint proteins at yeast centrosomes" for consideration by *eLife*.. Your article has been favorably evaluated by Richard Losick as the Senior editor and two reviewers, one of whom, Yukiko Yamashita, is a member of our Board of Reviewing Editors.

The reviewers have discussed the reviews with one another and the Reviewing Editor has drafted this decision to help you prepare a revised submission.

Summary:

The reviewers felt that major concerns in the previous revisions were adequately addressed. But the reviewer #3 felt that the description can be improved (accuracy of interpreting results, discussion on alternative possibilities etc.). This can be all taken care by revising the text.

*Reviewer #1:*

I believe that the authors adequately addressed reviewers' comments in this revised version. In particular, the use of Spc72-GBP Bfa1-GFP clearly addressed the concern on the correlational nature of FRET experiments. Now, the results support their idea more strongly that Spc72-Bfa1 interaction represents a critical step in SPOC.

*Reviewer #3:*

My main suggestion to the authors is that they elaborate on all of the implications of their experimental findings, and clearly discuss alternative models that are also consistent with the data.

1) Address the inconsistency regarding the proposed model of simultaneous interaction of one Bfa1 with both Nud1 and Spc72.

The authors use FRET between Spc72 C-terminus and Bfa1 C-terminus and in vitro pull-down to conclude that the two interact (Figure 1). However, the authors also find that the deletion of Spc72 does not have any detectable effect on steady-state Bfa1 recruitment or its turn-over rate at the SPB (Figure 4). The authors reconcile these findings by arguing that the same Bfa1 molecule simultaneously interacts with both Spc72 and Nud1.

If there are two binding sites in close physical proximity, then their combined affinity can be expected to be more than the sum of the individual affinities. Even if this expectation is somehow not relevant in this case, the deletion of one of the two binding sites (Spc72) should have a measurable effect on either the recruitment or turn-over of Bfa1. The in vivo data do not support this.

I can think of four possible reasons for these contradictory data: (1) a mutation/suppressor counteracts the effects of Spc72 deletion on Bfa1 recruitment, (2) the increased stoichiometry of Bfa1:Nud1 in the *spc72Δ* strain exactly compensates for the loss of one binding site, (3) the affinity of the Bfa1 binding site in Spc72 is very low, and Nud1 is mostly responsible for Bfa1 recruitment, and (4) the in vitro result is an artifact; Spc72 does not interact with Bfa1 in vivo. It should be noted that the Spc72-Bfa1 FRET is correlative. Changes in this FRET could simply reflect increased separation between the fluorophores due to the binding of Bmh1 to Bfa1.

The authors should clearly discuss the inconsistency in the in vivo and in vitro result and other possible interpretations of the data.

2) Discuss all possible reasons for the origin of asymmetric recruitment of Bfa1 by mother and daughter SPB.

The point discussed above also affects the authors' explanation for why the daughter SPB recruits higher amounts of Bfa1. The authors note that "Previously, correlative experiments excluded the possibility that an age-related difference of the two SPBs could account for the preferred binding of Bfa1-Bub2 complexes to the dSPB (Juanes et al., 2013; Pereira et al., 2001). However, these studies did not exclude the possibility that asymmetry arose from the association of Bfa1-Bub2 complexes to different molecules at the mother and daughter SPB."

If the authors' model that Spc72 is a Nud1-independent receptor of Bfa1 is true, then they should clarify that Spc72 itself is asymmetrically distributed at the mother and daughter SPB. In fact, the Juanes et al. study demonstrates that this asymmetry in Spc72 levels is responsible for orienting the daughter SPB.

On the other hand, if Spc72 does not provide a significant binding interface for Bfa1 as discussed in point #1, then the asymmetry could arise from differential phosphoregulation of Bfa1 at the mother and daughter SPBs.

3) Cite prior studies that demonstrate that the satisfaction of mitotic checkpoint locally produces a dominant signal that silences the checkpoint globally.

At least two prior studies demonstrate that it takes very little to tip the balance of the cell cycle biochemistry: (a) "Synchronous nuclear-envelope breakdown and anaphase onset in plant multinucleate cells" Protoplasma (2001) 218:192-202, and (b) "Mitosis in vertebrate somatic cells with two spindles: Implications for the metaphase/anaphase transition checkpoint and cleavage" PNAS 94, pp. 5107-5112. Yeast MEN appears to be the latest example of this intriguing aspect of the cell cycle.

---

## [Author Response]

Summarized review comments:

This is an interesting and important study that addresses the mechanism of SPOC (spindle orientation checkpoint) that ensures proper nuclear segregation prior to mitotic exit. SPOC is a critical checkpoint mechanism that regulates mitotic exit (i.e. upstream of mitotic exit network, MEN) depending on the state of spindle positioning within the cell to ensure correct segregation of two nuclei into two daughter cells. An elegant series of FRET experiments are used to assess the association of Bfa1 with the SPB components Nud1 and Spc72. The data show that Bfa1 simultaneously associates with Spc72 and Nud1 but that it is the association with Spc72 that alters in the SPOC dependent manner.

The reviewers agreed that this study provides a significant progress toward the understanding of molecular mechanism for SPOC. The novel role of Spc72 as a scaffold for Cdc5 and Kin4 is the key point of this study.

In this regard, the major concern was that FRET data are still correlational because the functional significance of Spc72 interactions (measured by FRET) with SPOC components is not defined. So, this correlational nature has to be mentioned in the text.

In the most rigorous manner, one could 1) map the interaction domains of Spc72 that recruits Cdc5 and Kin4, then 2) generate separation of function alleles to define the phenotypic outcome, which will provide a fine model on how Spc72 operates in SPOC signaling, and 3) use FKBP-FRB system to test the model by manipulating Spc72's interaction with SPOC components.

However, the reviewers also agreed that the above points 1 and 2 may be beyond the scope of the current work, and that the existing data presented in this manuscript are sufficient to formulate a working model (as presented in Figure 6). Therefore, the reviewers agreed that the authors should describe their model of how the differential change in Bfa1-Spc72 and Bfa1-Nud1 FRET relates to the ability of Cdc5 and Kin4 to phosphorylate Bfa1, and test the model by additional, relatively simple experiment(s) along the line of above point 3).

We thank the reviewers for the suggestion. The correlative nature of FRET data is now mentioned in the text (subsection “SPOC activation disturbs Bfa1-Spc72 interaction through Kin4/Bmh1 branch of the SPOC”, last paragraph). We now performed additional experiments to test the functional significance of the change in Bfa1-Spc72 FRET and also to test the model that dissociation of Bfa1 from Spc72 prevents inhibitory phosphorylation of Bfa1 by Cdc5. As suggested by the reviewers, we manipulated Spc72-Bfa1 interaction so that Bfa1 remained constitutively bound to Spc72. For this, we expressed *BFA1-GFP* in cells carrying Spc72 fused to the GFP-binding protein (GBP). In the presence of Spc72-GBP, Bfa1-GFP localized at both SPBs independently of the spindle position. We show that *BFA1-GFP SPC72-GBP* cells were SPOC deficient, indicating the functional significance of remodeling Bfa1-Spc72 interaction upon spindle misalignment. We also show that Bfa1-GFP became hyperphosphorylated in nocodazole-treated *SPC72-GBP* but not in *SPC72* cells. Depletion of *CDC5* greatly reduced the Bfa1-GFP hyperphosphorylation in *SPC72-GBP* cells, establishing the contribution of Bfa1-Spc72 interaction to Cdc5 phosphorylation of Bfa1. *KIN4* also phosphorylated Bfa1-GFP in *SPC72-GBP* cells. These data indicate that Kin4 and Cdc5 phosphorylate Bfa1 upon SPOC activation, if Bfa1 remains constitutively bound to Spc72. These new data support our model and are now shown in Figure 7, explained in the subsection “Bfa1-Spc72 interaction enhances Bfa1 phosphorylation by Cdc5 and Kin4” and discussed in the subsection “The γ-tubulin complex receptor Spc72 coordinates activation and inhibition of Bfa1- Bub2”. We also described the model in the legend to Figure 7.

[Editors' note: further revisions were requested prior to acceptance, as described below.]

Reviewer #3:

My main suggestion to the authors is that they elaborate on all of the implications of their experimental findings, and clearly discuss alternative models that are also consistent with the data.

1) Address the inconsistency regarding the proposed model of simultaneous interaction of one Bfa1 with both Nud1 and Spc72.

The authors use FRET between Spc72 C-terminus and Bfa1 C-terminus and in vitro pull-down to conclude that the two interact (Figure 1). However, the authors also find that the deletion of Spc72 does not have any detectable effect on steady-state Bfa1 recruitment or its turn-over rate at the SPB (Figure 4). The authors reconcile these findings by arguing that the same Bfa1 molecule simultaneously interacts with both Spc72 and Nud1.

If there are two binding sites in close physical proximity, then their combined affinity can be expected to be more than the sum of the individual affinities. Even if this expectation is somehow not relevant in this case, the deletion of one of the two binding sites (Spc72) should have a measurable effect on either the recruitment or turn-over of Bfa1. The in vivo data do not support this.

I can think of four possible reasons for these contradictory data: (1) a mutation/suppressor counteracts the effects of Spc72 deletion on Bfa1 recruitment, (2) the increased stoichiometry of Bfa1:Nud1 in the spc72Δ strain exactly compensates for the loss of one binding site, (3) the affinity of the Bfa1 binding site in Spc72 is very low, and Nud1 is mostly responsible for Bfa1 recruitment, and (4) the in vitro result is an artifact; Spc72 does not interact with Bfa1 in vivo. It should be noted that the Spc72-Bfa1 FRET is correlative. Changes in this FRET could simply reflect increased separation between the fluorophores due to the binding of Bmh1 to Bfa1.

The authors should clearly discuss the inconsistency in the in vivo and in vitro result and other possible interpretations of the data.

We now added a new section in the Discussion (“Contribution of Nud1 and Spc72 for Bfa1 SPB binding behaviour”). In this section, we discuss whether Bfa1 binds to Nud1 and Spc72 separately or simultaneously. We discuss our in vitro and FRET data as well as the data obtained from the analysis of Bfa1 SPB binding behaviour in *spc72∆* cells. We also discuss the inconsistencies of in vitro and in vivo data (in respect to Spc72’s contribution to Bfa1 SPB binding behaviour):

“Contribution of Nud1 and Spc72 for Bfa1 SPB binding behavior:

Biochemical and FRET data are consistent with the binding of Bfa1 to Nud1 and Spc72. This raises the question of how Bfa1 associates with these proteins at the SPB. Bfa1 could bind separately or simultaneously to Nud1 and Spc72. […] In any case, further biochemical and biophysical studies will be necessary to evaluate the affinity of Bfa1 towards Nud1 and Spc72, and to establish whether the same Bfa1 molecule can bind simultaneously to Nud1 and Spc72.”

We now discuss that loss of FRET could also indicate separation of the fluorophores only:

“We observed loss of Bfa1-Spc72 (but not Bfa1-Nud1) FRET upon SPOC activation. The loss of FRET signal between Bfa1 and Spc72 might indicate the physical separation of the two proteins or only the separation of the fluorophores coupled to Bfa1 and Spc72 C-termini. […] Such regulation may resemble the conformational change that Mad2 undergoes upon activation of the spindle assembly checkpoint (SAC) (Luo et al., 2002; Luo et al., 2004; Mapelli et al., 2007; Mapelli and Musacchio, 2007; Sironi et al., 2002; Yang et al., 2007).”

2) Discuss all possible reasons for the origin of asymmetric recruitment of Bfa1 by mother and daughter SPB.

The point discussed above also affects the authors' explanation for why the daughter SPB recruits higher amounts of Bfa1. The authors note that "Previously, correlative experiments excluded the possibility that an age-related difference of the two SPBs could account for the preferred binding of Bfa1-Bub2 complexes to the dSPB (Juanes et al., 2013; Pereira et al., 2001). However, these studies did not exclude the possibility that asymmetry arose from the association of Bfa1-Bub2 complexes to different molecules at the mother and daughter SPB."

If the authors' model that Spc72 is a Nud1-independent receptor of Bfa1 is true, then they should clarify that Spc72 itself is asymmetrically distributed at the mother and daughter SPB. In fact, the Juanes et al. study demonstrates that this asymmetry in Spc72 levels is responsible for orienting the daughter SPB.

On the other hand, if Spc72 does not provide a significant binding interface for Bfa1 as discussed in point #1, then the asymmetry could arise from differential phosphoregulation of Bfa1 at the mother and daughter SPBs.

We expand the section about the asymmetric Bfa1 SPB binding to discuss the role of post-translational modifications and SPB age (including Spc72 asymmetry):

“Asymmetric association of Bfa1 with SPBs in cells with correctly aligned spindles:

The Bfa1-Bub2 complex is recruited preferentially to the dSPB (asymmetric binding) in cells progressing normally through the cell cycle. […] How cell polarity determinants control Bfa1 asymmetry is unclear. Daughter cell associated factors could stabilize the Bfa1-Nud1 interaction at the dSPB, for example by influencing the post-translational regulation of Bfa1-Bub2 or Nud1.”

3) Cite prior studies that demonstrate that the satisfaction of mitotic checkpoint locally produces a dominant signal that silences the checkpoint globally.

At least two prior studies demonstrate that it takes very little to tip the balance of the cell cycle biochemistry: (a) "Synchronous nuclear-envelope breakdown and anaphase onset in plant multinucleate cells" Protoplasma (2001) 218:192-202, and (b) "Mitosis in vertebrate somatic cells with two spindles: Implications for the metaphase/anaphase transition checkpoint and cleavage" PNAS 94, pp. 5107-5112. Yeast MEN appears to be the latest example of this intriguing aspect of the cell cycle.

We thank the reviewer for this suggestion. We now discuss previous work showing checkpoint behaviour in binucleated cells:

“Of importance, our data indicate that the mitotic exit stop-signal generated by the SPOC at the misaligned spindle cannot sustain the mitotic arrest if the checkpoint is satisfied in the neighboring spindle. […] We postulate that misbalanced checkpoint integrity in binucleated cells might be a driver for genome instability and cancer development.”